# SCORE-BASED VARIATIONAL INFERENCE FOR INVERSE PROBLEMS

## ABSTRACT

Existing diffusion-based methods for inverse problems sample from the posterior using score functions and accept the generated random samples as solutions. In applications that posterior mean is preferred, we have to generate multiple samples from the posterior which is time-consuming. In this work, by analyzing the probability density evolution of the conditional reverse diffusion process, we prove that the posterior mean can be achieved by tracking the mean of each reverse diffusion step. Based on that, we establish a framework termed reverse mean propagation (RMP) that targets the posterior mean directly. We show that RMP can be implemented by solving a variational inference problem, which can be further decomposed as minimizing a reverse KL divergence at each reverse step. We further develop an algorithm that optimizes the reverse KL divergence with natural gradient descent using score functions and propagates the mean at each reverse step. Experiments demonstrate the validity of the theory of our framework and show that our algorithm outperforms state-of-the-art algorithms on reconstruction performance with lower computational complexity in various inverse problems.

## 1 INTRODUCTION

Diffusion models (Sohl-Dickstein et al., 2015; Song & Ermon, 2019; Ho et al., 2020; Song et al., 2020a; Rombach et al., 2022) have shown impressive performance for image generation. For diffusion models such as diffusion denoising probability model (DDPM) (Ho et al., 2020) and denoising score matching with Langevin dynamics (SMLD) (Song & Ermon, 2019), the essential part is the learning of score functions of data distributions with large datasets. By approximating score functions with neural networks such as U-Net (Ronneberger et al., 2015; Song & Ermon, 2019), the prior of complex data distributions can be learned implicitly which encourages many applications. Inverse problems aim to recover an unknown state $x_0$ from observation $y$, which is fundamental to various research areas such as wireless communication, image processing and natural language processing. Recent works (Jalal et al., 2020; Kawar et al., 2021; Song et al., 2020a; 2021; Chung et al., 2022b; Kawar et al., 2022; Meng & Kabashima, 2022; Chung et al., 2022a; Laumont et al., 2022; Mardani et al., 2023) have shown that diffusion models can be used for solving inverse problems since the prior of data distribution is learned implicitly with score functions and score-based priors are more efficient to train (Song & Ermon, 2020).

Based on Bayes' rule, diffusion models are used for the generation of data from the posterior distribution with score functions of data and likelihood, and thus can be applied in solving reverse problems. The main difficulty of applying diffusion models to solving inverse problems is the calculation of likelihood score. SNIPS (Kawar et al., 2021) and DDRM (Kawar et al., 2022) are proposed to solve noisy linear inverse problems with diffusion process in the spectral domain. In these methods, the measurement and data to be estimated are transformed into the spectral domain via singular value decomposition (SVD), and the conditional score can be calculated with SVD explicitly. In Meng & Kabashima (2022), the authors propose to approximate the likelihood score by a noise-perturbed pseudo-likelihood score which has a closed form under certain assumptions and can be efficiently calculated using SVD for noisy linear inverse problems. MCG (Chung et al., 2022b) circumvents the calculation of likelihood score by projections onto the measurement constrained manifold. In DPS (Chung et al., 2022a), the Laplace method is used for the approximation of the likelihood score for general inverse problems. Instead of directly approximating the likelihood

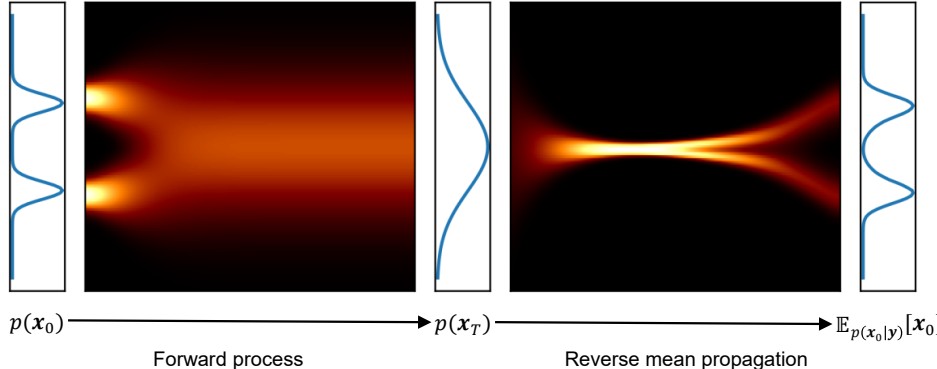

Figure 1: An illustration of RMP for Gaussian mixture model. In the experiment, the data prior is $p(x_0) = \frac{1}{2}(\mathcal{N}(x_0; \mu_1, v_1^2) + \mathcal{N}(x_0; \mu_2, v_2^2))$ and measurement $y = ax + v_0\varepsilon$ where $\mu_1 = -1$, $\mu_2 = 1$, $v_1 = v_2 = 0.2$, $v_0 = 0.5$, $a = 1$, $\varepsilon \sim \mathcal{N}(0, 1)$ and $T = 1000$. RMP is deterministic when $y$ and $\boldsymbol{x}_T$ are given and the final output converges to $\mathbb{E}_{p(x_0|y)}[x_0]$.

score, a variational inference based method termed RED-Diff (Mardani et al., 2023) optimizes the reconstruction loss with score matching regularization.

The Fokker-Planck equation (Risken, 1996; Jordan et al., 1998) describes the evolution of the probability density of a stochastic process. In this work, we present an evolution analysis of the probability density for the conditional reverse diffusion process, as described by the Fokker-Planck equation. Based on the analysis, we propose a variational inference framework that minimizes the reverse KL divergence, referred to as Reverse Mean Propagation (RMP), since it propagates the mean at each reverse step, as illustrated in Fig. 1. Different from the previous works that draw samples from the posterior with a reverse diffusion process, our proposed RMP algorithm is essentially an approximation of the Fokker-Planck equation for a conditional reverse process that tracks the evolution of the probability density which is deterministic. This represents a fundamental difference between our method and previous methods. Unlike sampling-based methods that have to sample from the posterior multiple times and average out to get the posterior mean which is time-consuming, RMP converges to the posterior mean, i.e. the MMSE estimation in Bayesian inference, which is preferred in many applications (Kay, 1993). RMP simplifies the application of variational inference to a broader range of problems in a plug-and-play manner by adopting a score-based prior that can be learned using neural networks (Ho et al., 2020; Song et al., 2020b) and performs effectively when the score networks are available. This is contrary to traditional variational inference methods (Maestrini et al., 2025; Wand, 2017) for inverse problems that require complex models of prior. The main contributions of our work are summarized as follows:

- We characterize the evolution of the transaction probability density of the conditional reverse diffusion process in terms of mean and covariance, based on which we propose the RMP framework that approaches the posterior mean.

- We connect variational inference with conditional diffusion process and propose an implementation of the RMP framework based on stochastic natural gradient descent, incorporating a score-based prior and suitable approximations that reduces complexity.

- We conduct extensive experiments to demonstrate the validity of the RMP framework theory and show that RMP outperforms state-of-the-art algorithms in reconstruction performance with lower computational complexity across various problems.

## 2 BACKGROUND

### 2.1 INVERSE PROBLEMS

An inverse problem is defined as the estimation of an unknown state or a latent $\boldsymbol{x}_0 \in \mathbb{R}^{N \times 1}$ from measurement $\boldsymbol{y} \in \mathbb{R}^{M \times 1}$. Specifically, the measurement process can be described by a measurement

operator $\mathcal{A}: \mathbb{R}^{N \times 1} \to \mathbb{R}^{M \times 1}$, and the final output is a noisy version of the measurement:

$$\boldsymbol{y} = \mathcal{A}(\boldsymbol{x}_0) + \boldsymbol{w}_0 \tag{1}$$

where $\boldsymbol{w}_0 \in \mathbb{R}^{M \times 1}$ is the measurement noise. Usually the measurement noise is assumed to be zero mean Gaussian with variance $\epsilon^2 \boldsymbol{I}$. Other noise models may also apply. In linear inverse problems, the measurement function $\mathcal{A}$ is linear and can be represented by a linear transform $\boldsymbol{A} \in \mathbb{R}^{M \times N}$.

## 2.2 VARIATIONAL INFERENCE

The inverse problem (1) can be formulated as a Bayesian estimation problem. The posterior distribution of $\boldsymbol{x}_0$ is given by $p(\boldsymbol{x}_0|\boldsymbol{y}) = \frac{p(\boldsymbol{y}|\boldsymbol{x}_0)p(\boldsymbol{x}_0)}{p(\boldsymbol{y})}$, where $p(\boldsymbol{x}_0)$ is the prior of $\boldsymbol{x}_0$ and $p(\boldsymbol{y}|\boldsymbol{x}_0)$ is the conditional distribution of $\boldsymbol{y}$ given $\boldsymbol{x}_0$. The posterior mean, i.e., the MMSE estimator can be employed for the estimation of $\boldsymbol{x}_0$. However, the posterior $p(\boldsymbol{x}_0|\boldsymbol{y})$ is intractable in general since the prior $p(\boldsymbol{x}_0)$ and the likelihood $p(\boldsymbol{y}|\boldsymbol{x}_0)$ may be very complicated in real applications. An alternative way is to find an approximation of the posterior as in variational inference (VI). VI introduces a distribution $q_\phi(\boldsymbol{x}_0|\boldsymbol{y})$ and maximizes a lower bound of the log probability of marginal $p(\boldsymbol{y})$:

$$
\begin{aligned}
\log p(\boldsymbol{y}) &= \log \int \frac{q_\phi(\boldsymbol{x}_0|\boldsymbol{y})}{q_\phi(\boldsymbol{x}_0|\boldsymbol{y})} p(\boldsymbol{y}|\boldsymbol{x}_0)p(\boldsymbol{x}_0)d\boldsymbol{x}_0 \\
&\geq \int q_\phi(\boldsymbol{x}_0|\boldsymbol{y}) \log \frac{p(\boldsymbol{y}, \boldsymbol{x}_0)}{q_\phi(\boldsymbol{x}_0|\boldsymbol{y})} d\boldsymbol{x}_0 = -\mathcal{F}_\phi(\boldsymbol{y}) \\
&= -\mathrm{KL}(q_\phi(\boldsymbol{x}_0|\boldsymbol{y})||p(\boldsymbol{x}_0|\boldsymbol{y})) + \log p(\boldsymbol{y})
\end{aligned} \tag{2}
$$

where the inequality is obtained by using the Jensen's inequality. The lower bound is referred as the evidence lower bound (ELBO) or the negative of free energy $\mathcal{F}(\boldsymbol{y})$. It is worth noting that maximizing ELBO is equivalent to minimizing the KL divergence between $q_\phi(\boldsymbol{x}_0|\boldsymbol{y})$ and $p(\boldsymbol{x}_0|\boldsymbol{y})$ as shown in the last line of (2). Many methods have been proposed for the optimization of (2) such as mean field VI (Blei et al., 2017), black box VI (Ranganath et al., 2014), stochastic VI (Kingma & Welling, 2013) and normalizing flow VI (Rezende & Mohamed, 2015). However, these methods are difficult to be applied to real applications since prior $p(\boldsymbol{x}_0)$ is complicated and is usually learned by neural networks. It is shown that learning the distribution of high dimensional data through score matching (Vincent, 2011) directly is inaccurate since the existance of low density data regions (Song & Ermon, 2019). Also, perturbing data with Gaussian noise makes the data distribution more amenable to learn (Song & Ermon, 2019) which is the core of score-based generative model.

## 2.3 SCORE-BASED GENERATIVE MODELS

Score-based generative models or diffusion models generate samples of a data distribution from the reverse process of a diffusion process. The diffusion process is also called the forward process where Gaussian noise is added gradually to the original data distribution until the noisy data are approximately Gaussian-distributed. More specifically, the diffusion process is a Markov process with joint probability of its latent states $\{\boldsymbol{x}_k\}_{k=0}^T$ given by

$$p(\boldsymbol{x}_{T:0}) = p(\boldsymbol{x}_0) \prod_{k=0}^{T-1} p(\boldsymbol{x}_{k+1}|\boldsymbol{x}_k). \tag{3}$$

Two classes of widely studied diffusion models, i.e., the variance preserving (VP) diffusion model (Ho et al., 2020) and the variance exploding (VE) diffusion model (Song & Ermon, 2019) are distinguished by Markov diffusion kernel $p(\boldsymbol{x}_{k+1}|\boldsymbol{x}_k)$. For VE diffusion, $p(\boldsymbol{x}_{k+1}|\boldsymbol{x}_k) = \mathcal{N}(\boldsymbol{x}_{k+1}; \boldsymbol{x}_k, (\sigma_{k+1}^2 - \sigma_k^2)\boldsymbol{I})$, where $\sigma_T^2 > \sigma_{T-1}^2 > \cdots > \sigma_1^2 > \sigma_0^2 = 0$, and for VP diffusion, $p(\boldsymbol{x}_{k+1}|\boldsymbol{x}_k) = \mathcal{N}(\boldsymbol{x}_{k+1}; \sqrt{1 - \beta_{k+1}}\boldsymbol{x}_k, \beta_{k+1}\boldsymbol{I})$ where $\beta_T > \beta_{T-1} > \cdots > \beta_1 > \beta_0 = 0$. The learning of score function $\nabla_{\boldsymbol{x}_k} \log p(\boldsymbol{x}_k)$ is essential to the sample generation for both diffusion models, and thus such models are called score-based generative models. In VP diffusion models, a variational reverse process is leaned to minimize the KL divergence between forward and reverse process. The score functions of perturbed data distributions are trained with a variational bound. Samples are generated from the learned reverse process with ancestral sampling method. In VE diffusion models, the score functions of perturbed data distributions are learned with a score network using denosing score matching (Vincent, 2011). Samples are generated with Langevin dynamics method. It is worth noting that, similar to VE diffusion, the training of VP diffusion is also equivalent to learning the score functions of perturbed data distributions (Song et al., 2020b).

## 3 DIFFUSION PROCESS AND POSTERIOR ESTIMATION

The measurement $\boldsymbol{y}$ in (1) and diffusion states $\{\boldsymbol{x}_k\}_{k=0}^T$ in (3) form a new Markov chain $\boldsymbol{y} \rightarrow \boldsymbol{x}_0 \rightarrow \boldsymbol{x}_1 \cdots \rightarrow \boldsymbol{x}_T$ and the reverse conditional $p_k(\boldsymbol{x}_k|\boldsymbol{x}_{k+1}, \boldsymbol{y})$ is given by $p_k(\boldsymbol{x}_k|\boldsymbol{x}_{k+1}, \boldsymbol{y}) = \frac{p(\boldsymbol{x}_{k+1}|\boldsymbol{x}_k)p(\boldsymbol{x}_k|\boldsymbol{y})}{p(\boldsymbol{x}_{k+1}|\boldsymbol{y})}, \forall k = 0, \cdots T-1$. In this part, we focus on the property of reverse conditional $p_k(\boldsymbol{x}_k|\boldsymbol{x}_{k+1}, \boldsymbol{y})$. We relate the Markov chain $\{\boldsymbol{x}_k\}_{k=0}^T$ to continuous stochastic process $\{\boldsymbol{x}_t\}_{t=0}^1$ by letting $\boldsymbol{x}_k = \boldsymbol{x}_{t=k\Delta t}$, where $\Delta t = \frac{1}{T}$. Then, the discrete diffusion process (3) becomes a continuous process in the limit $\Delta t \rightarrow 0$. We have the following results in the limit of $\Delta t \rightarrow 0$.

**Proposition 1** *For diffusion models with forward process (3), the reverse conditional $p_k(\boldsymbol{x}_k|\boldsymbol{x}_{k+1}, \boldsymbol{y}), \forall k = 0, \cdots, T-1$, is Gaussian when $\Delta t \rightarrow 0$. For VE and VP diffusion, the mean and covariance of $p_k(\boldsymbol{x}_k|\boldsymbol{x}_{k+1}, \boldsymbol{y})$ are tractable with mean given by*

$$\boldsymbol{\mu}_k(\boldsymbol{x}_{k+1}, \boldsymbol{y}) = \boldsymbol{V}_{k,1}\boldsymbol{x}_{k+1} + \boldsymbol{V}_{k,2}\mathbb{E}_{p(\boldsymbol{x}_0|\boldsymbol{y})}[\boldsymbol{x}_0] \qquad (4)$$

*where $\boldsymbol{V}_{k,1} = (\sigma_k^2\boldsymbol{I} + \boldsymbol{C}_{\boldsymbol{x}_0})(\sigma_{k+1}^2\boldsymbol{I} + \boldsymbol{C}_{\boldsymbol{x}_0})^{-1}$ and $\boldsymbol{V}_{k,2} = (\sigma_{k+1}^2 - \sigma_k^2)(\sigma_{k+1}^2\boldsymbol{I} + \boldsymbol{C}_{\boldsymbol{x}_0})^{-1}$ for VE diffusion, and $\boldsymbol{V}_{k,1} = \sqrt{\alpha_{k+1}}((1-\bar{\alpha}_k)\boldsymbol{I} + \bar{\alpha}_k\boldsymbol{C}_{\boldsymbol{x}_0})((1-\bar{\alpha}_{k+1})\boldsymbol{I} + \bar{\alpha}_{k+1}\boldsymbol{C}_{\boldsymbol{x}_0})^{-1}$ and $\boldsymbol{V}_{k,2} = \sqrt{\bar{\alpha}_k}(1-\alpha_{k+1})((1-\bar{\alpha}_{k+1})\boldsymbol{I} + \bar{\alpha}_{k+1}\boldsymbol{C}_{\boldsymbol{x}_0})^{-1}$ for VP diffusion. $\mathbb{E}_{p(\boldsymbol{x}_0|\boldsymbol{y})}[\boldsymbol{x}_0]$ and $\boldsymbol{C}_{\boldsymbol{x}_0}$ are the mean and covariance of $p(\boldsymbol{x}_0|\boldsymbol{y})$ respectively. $\bar{\alpha}_k = \prod_{i=0}^k \alpha_i$, $\alpha_i = 1 - \beta_i$. The covariance of $p_k(\boldsymbol{x}_k|\boldsymbol{x}_{k+1}, \boldsymbol{y})$ for VE and VP diffusion models are given respectively by*

$$\boldsymbol{C}_{k,VE} = (\sigma_{k+1}^2 - \sigma_k^2)(\sigma_k^2\boldsymbol{I} + \boldsymbol{C}_{\boldsymbol{x}_0})(\sigma_{k+1}^2\boldsymbol{I} + \boldsymbol{C}_{\boldsymbol{x}_0})^{-1}$$

$$\boldsymbol{C}_{k,VP} = \frac{\beta_{k+1}}{1 - \beta_{k+1}}((1-\bar{\alpha}_k)\boldsymbol{I} + \bar{\alpha}_k\boldsymbol{C}_{\boldsymbol{x}_0})\left(\left(\frac{\beta_{k+1}}{1 - \beta_{k+1}} + 1 - \bar{\alpha}_k\right)\boldsymbol{I} + \bar{\alpha}_k\boldsymbol{C}_{\boldsymbol{x}_0}\right)^{-1}. \qquad (5)$$

Proposition 1 generalizes the Gaussian property of $p_t(\boldsymbol{x}_t|\boldsymbol{x}_{t+\Delta t})$ to the conditional case $p_t(\boldsymbol{x}_t|\boldsymbol{x}_{t+\Delta t}, \boldsymbol{y})$ when $\Delta t \rightarrow 0$. The essential is that the reverse process can also be expressed by a reverse SDE (Song et al., 2020b) and the evolution of transaction probability $p_k(\boldsymbol{x}_k|\boldsymbol{x}_{k+1}, \boldsymbol{y})$ can be described by the Kolmogorov backward equation in the proof of Fokker-Planck equation Risken (1996); Jordan et al. (1998). Based on Proposition 1, we obtain the following main result.

**Definition 1** *The reverse mean propagation chain of a diffusion process is defined as*

$$\boldsymbol{\mu}_T \rightarrow \boldsymbol{\mu}_{T-1}(\boldsymbol{x}_T = \boldsymbol{\mu}_T, \boldsymbol{y}) \rightarrow \cdots \rightarrow \boldsymbol{\mu}_1(\boldsymbol{x}_2 = \boldsymbol{\mu}_2, \boldsymbol{y}) \rightarrow \boldsymbol{\mu}_0(\boldsymbol{x}_1 = \boldsymbol{\mu}_1, \boldsymbol{y}) \qquad (6)$$

*where $\boldsymbol{\mu}_k(\boldsymbol{x}_{k+1} = \boldsymbol{\mu}_{k+1}, \boldsymbol{y})$ is the mean of $p_k(\boldsymbol{x}_k|\boldsymbol{x}_{k+1} = \boldsymbol{\mu}_{k+1}, \boldsymbol{y}), \forall k = 0, \cdots, T-1$, and $\boldsymbol{\mu}_T$ and $\boldsymbol{\mu}_0$ are the initial point and the end point of the reverse chain respectively.*

**Theorem 1** *For VE diffusion, when $\Delta t \rightarrow 0$, the end point of the reverse chain, i.e., $\boldsymbol{\mu}_0$ is given by*

$$\boldsymbol{\mu}_0 = (\sigma_0^2\boldsymbol{I} + \boldsymbol{C}_{\boldsymbol{x}_0})(\sigma_T^2\boldsymbol{I} + \boldsymbol{C}_{\boldsymbol{x}_0})^{-1}\boldsymbol{\mu}_T + (\sigma_T^2 - \sigma_0^2)(\sigma_T^2\boldsymbol{I} + \boldsymbol{C}_{\boldsymbol{x}_0})^{-1}\mathbb{E}_{p(\boldsymbol{x}_0|\boldsymbol{y})}[\boldsymbol{x}_0] \qquad (7)$$

*and $\boldsymbol{\mu}_0 \rightarrow \mathbb{E}_{p(\boldsymbol{x}_0|\boldsymbol{y})}[\boldsymbol{x}_0]$ as $\sigma_T \rightarrow \infty$. For VP diffusion, when $\Delta t \rightarrow 0$, $\boldsymbol{\mu}_0$ is given by*

$$\begin{aligned} \boldsymbol{\mu}_0 = &\sqrt{\bar{\alpha}_T}((1-\bar{\alpha}_0)\boldsymbol{I} + \bar{\alpha}_0\boldsymbol{C}_{\boldsymbol{x}_0})((1-\bar{\alpha}_T)\boldsymbol{I} + \bar{\alpha}_T\boldsymbol{C}_{\boldsymbol{x}_0})^{-1}\boldsymbol{\mu}_T \\ &+ (1-\bar{\alpha}_T)((1-\bar{\alpha}_T)\boldsymbol{I} + \bar{\alpha}_T\boldsymbol{C}_{\boldsymbol{x}_0})^{-1}\mathbb{E}_{p(\boldsymbol{x}_0|\boldsymbol{y})}[\boldsymbol{x}_0] \end{aligned} \qquad (8)$$

*and $\boldsymbol{\mu}_0 \rightarrow \mathbb{E}_{p(\boldsymbol{x}_0|\boldsymbol{y})}[\boldsymbol{x}_0]$ as $\bar{\alpha}_T \rightarrow 0$.*

According to Theorem 1, the posterior mean can be obtained by tracking the mean at each reverse step. By calculating the reverse mean and following the reverse chain in (6), we get a posterior estimation framework termed Reverse Mean Propagation (RMP) for inverse problems as presented in Algorithm 1. We note that when the initial point of the reverse chain $\boldsymbol{\mu}_T$ and $\boldsymbol{y}$ are given, the reverse chain is deterministic and converges to the posterior mean $\mathbb{E}_{p(\boldsymbol{x}_0|\boldsymbol{y})}[\boldsymbol{x}_0]$.

## 4 SCORE-BASED VARIATIONAL INFERENCE

In this section, we propose a score-based variational inference method to implement the RMP framework. We show that tracking the mean of the reverse process of each step can be formulated as a sequential of variational inference problems which we prove to be equivalent to a variational inference problem for all latent variables. We solve the variational inference by stochastic natural gradient descent with approximations that simplify the calculation.

---

**Algorithm 1:** Reverse Mean Propagation (RMP)

**Input** : $\boldsymbol{y}, T, \boldsymbol{\mu}_T$
1 **for** $k = T - 1 : 0$ **do**
2 $\quad$ Propagate the reverse mean: $\boldsymbol{x}_{k+1} = \boldsymbol{\mu}_{k+1}$
3 $\quad$ Calculate the reverse mean of $p_k$: $\boldsymbol{\mu}_k(\boldsymbol{x}_{k+1} = \boldsymbol{\mu}_{k+1}, \boldsymbol{y}) = \mathbb{E}_{p_k(\boldsymbol{x}_k | \boldsymbol{x}_{k+1} = \boldsymbol{\mu}_{k+1}, \boldsymbol{y})}[\boldsymbol{x}_k]$
4 **end**
**Output** : $\boldsymbol{\mu}_0$

---

### 4.1 RMP AS VARIATIONAL INFERENCE

In Section 3, we present the RMP framework based on the reverse diffusion process. However, in practice, the reverse mean in RMP cannot be calculated using (4) since $\mathbb{E}_{p(\boldsymbol{x}_0 | \boldsymbol{y})}[\boldsymbol{x}_0]$ and $\boldsymbol{C}_{\boldsymbol{x}_0}$ are unknown. We now show that the RMP framework can be implemented using variational inference. Instead of applying variational inference on the conditional posterior of $\boldsymbol{x}_0$ as in (2), we focus on the joint conditional posterior of $\{\boldsymbol{x}_k\}_{k=0}^T$, i.e., $p(\boldsymbol{x}_{0:T} | \boldsymbol{y})$, which includes all the latent variables in the diffusion process. The variational reverse process with joint conditional is defined by

$$q(\boldsymbol{x}_{0:T} | \boldsymbol{y}) = q(\boldsymbol{x}_T | \boldsymbol{y}) \prod_{k=0}^{T-1} q_k(\boldsymbol{x}_k | \boldsymbol{x}_{k+1}, \boldsymbol{y}), \qquad (9)$$

where $q_k(\boldsymbol{x}_k | \boldsymbol{x}_{k+1}, \boldsymbol{y}) = \mathcal{N}(\boldsymbol{x}_k; \boldsymbol{\mu}_k(\boldsymbol{x}_{k+1}, \boldsymbol{y}), \boldsymbol{C}_k(\boldsymbol{x}_{k+1}, \boldsymbol{y})), \forall k = 0 : T - 1$. We set $q(\boldsymbol{x}_T | \boldsymbol{y}) = \mathcal{N}(\boldsymbol{x}_T; 0, \boldsymbol{I})$. The variational reverse process is chosen as a Markov chain since the reverse of the diffusion process (3) is a Markov process. The KL divergence between variational joint posterior $q(\boldsymbol{x}_{0:T} | \boldsymbol{y})$ and and joint posterior $p(\boldsymbol{x}_{0:T} | \boldsymbol{y})$ is given by

$$\text{KL}(q || p) = \int q(\boldsymbol{x}_{0:T} | \boldsymbol{y}) \log \frac{q(\boldsymbol{x}_{0:T} | \boldsymbol{y})}{p(\boldsymbol{x}_{0:T} | \boldsymbol{y})} d\boldsymbol{x}_{0:T} \qquad (10)$$

where the forward joint posterior $p(\boldsymbol{x}_{0:T} | \boldsymbol{y}) = p(\boldsymbol{x}_T | \boldsymbol{y}) \prod_{k=0}^{T-1} p(\boldsymbol{x}_k | \boldsymbol{x}_{k+1}, \boldsymbol{y})$. For VE diffusion $p(\boldsymbol{x}_T | \boldsymbol{y}) = \mathcal{N}(\boldsymbol{x}_T; 0, \sigma_T^2 \boldsymbol{I})$ and for VP diffusion $p(\boldsymbol{x}_T | \boldsymbol{y}) = \mathcal{N}(\boldsymbol{x}_T; 0, \boldsymbol{I})$. The following proposition simplifies the minimization of $\text{KL}(q || p)$ with proof given in Appendix C.

**Proposition 2** *For a diffusion process with forward process (3), the KL divergence between variational $q(\boldsymbol{x}_{0:T} | \boldsymbol{y})$ and joint posterior $p(\boldsymbol{x}_{0:T} | \boldsymbol{y})$ defined in (10) equals*

$$KL(q || p) = \sum_{k=T-1}^{0} \int q(\boldsymbol{x}_{k+1} | \boldsymbol{y}) \int q_k(\boldsymbol{x}_k | \boldsymbol{x}_{k+1}, \boldsymbol{y}) \log \frac{q_k(\boldsymbol{x}_k | \boldsymbol{x}_{k+1}, \boldsymbol{y})}{p_k(\boldsymbol{x}_k | \boldsymbol{x}_{k+1}, \boldsymbol{y})} d\boldsymbol{x}_k d\boldsymbol{x}_{k+1}, \qquad (11)$$

*and the minimization of $KL(q || p)$ is equivalent to the minimization of*

$$KL(q_k || p_k) = \int q_k(\boldsymbol{x}_k | \boldsymbol{x}_{k+1}, \boldsymbol{y}) \log \frac{q_k(\boldsymbol{x}_k | \boldsymbol{x}_{k+1}, \boldsymbol{y})}{p_k(\boldsymbol{x}_k | \boldsymbol{x}_{k+1}, \boldsymbol{y})} d\boldsymbol{x}_k, \forall k = 0, \cdots, T - 1. \qquad (12)$$

According to Proposition 2, we can minimize the KL divergence between $q$ and $p$ by minimizing the KL divergence $\text{KL}(q_k || p_k)$, i.e., solve the following VI problem at each reverse step $k$:

$$q_k^\star = \arg \min_{q_k} \text{KL}(q_k || p_k), \forall k = 0, \cdots, T - 1. \qquad (13)$$

By propagating the mean of $q_k$ and solving problem (13) at each reverse step $k$, we can approximate the RMP framework in Algorithm 1 based on variational inference, as detailed below.

### 4.2 VARIATIONAL INFERENCE BY NATURAL GRADIENT DESCENT

For $q_k(\boldsymbol{x}_k | \boldsymbol{x}_{k+1}, \boldsymbol{y}) = \mathcal{N}(\boldsymbol{x}_k; \boldsymbol{\mu}_k, \Lambda_k^{-1} \boldsymbol{I})$, the KL divergence between $q_k$ and $p_k$ is given by

$$\begin{aligned} \text{KL}(q_k || p_k) &= \int q_k(\boldsymbol{x}_k | \boldsymbol{x}_{k+1}, \boldsymbol{y}) \log \frac{q_k(\boldsymbol{x}_k | \boldsymbol{x}_{k+1}, \boldsymbol{y})}{p_k(\boldsymbol{x}_k | \boldsymbol{x}_{k+1}, \boldsymbol{y})} d\boldsymbol{x}_k \\ &= -\frac{N}{2} \log(2\pi / \Lambda_k) - \frac{N}{2} - \mathbb{E}_{q_k}[\log p_k(\boldsymbol{x}_k | \boldsymbol{x}_{k+1}, \boldsymbol{y})]. \end{aligned} \qquad (14)$$

A common practice to optimize $\text{KL}(q_k||p_k)$ is to update variational parameters $\phi_k = \{\boldsymbol{\mu}_k, \Lambda_k\}$ using mini-batch stochastic gradient descent which involves the calculation of $\nabla_{\phi_k}\text{KL}(q_k||p_k)$. Since $q_k$ is Gaussian, the gradient of $\mathbb{E}_{q_k}[\log p_k(\boldsymbol{x}_k|\boldsymbol{x}_{k+1}, \boldsymbol{y})]$ in (14) with respect to variational parameters $\phi_k = \{\boldsymbol{\mu}_k, \Lambda_k\}$ have simple forms (Opper & Archambeau, 2009) which are given by:

$$\nabla_{\boldsymbol{\mu}_k}\mathbb{E}_{q_k}[\log p_k(\boldsymbol{x}_k|\boldsymbol{x}_{k+1}, \boldsymbol{y})] = \mathbb{E}_{q_k}[\nabla_{\boldsymbol{x}_k}\log p_k(\boldsymbol{x}_k|\boldsymbol{x}_{k+1}, \boldsymbol{y})]$$
$$\nabla_{\Lambda_k}\mathbb{E}_{q_k}[\log p_k(\boldsymbol{x}_k|\boldsymbol{x}_{k+1}, \boldsymbol{y})] = -\frac{1}{2}\Lambda_k^{-2}\mathbb{E}_{q_k}[\text{Tr}(\nabla_{\boldsymbol{x}_k}^2\log p_k(\boldsymbol{x}_k|\boldsymbol{x}_{k+1}, \boldsymbol{y})]$$

(15)

where $\nabla_{\boldsymbol{x}_k}\log p_k(\boldsymbol{x}_k|\boldsymbol{x}_{k+1}, \boldsymbol{y})$ and $\nabla_{\boldsymbol{x}_k}^2\log p_k(\boldsymbol{x}_k|\boldsymbol{x}_{k+1}, \boldsymbol{y})$ are the gradient and Hessian of $\log p_k(\boldsymbol{x}_k|\boldsymbol{x}_{k+1}, \boldsymbol{y})$ respectively, and $\text{Tr}(\cdot)$ returns the trace of the input matrix.

As a special case of steepest descent, gradient descent updates parameter that lies in the Euclidean space. However, our objective is to optimize parameters that represent a distribution, it makes sense to take the steepest descent direction in the distribution space. As in natural gradient descent (Martens, 2020), the parameter to be optimized lies on a Riemannian manifold and we choose the steepest descent direction along that manifold. Thus, we choose KL-divergence as the metric of distribution space and take steepest descent in this space. For KL-divergence metric, the natural gradient of parameter of a loss function $\mathcal{L} = \mathbb{E}_{q_\phi}[h(\boldsymbol{x})]$ is defined as $\tilde{\nabla}_\phi\mathcal{L} = \boldsymbol{F}_\phi^{-1}\nabla_\phi\mathcal{L}$ (Martens, 2020), where $\boldsymbol{F}_\phi$ is the Fisher information matrix of $\phi$ given by the variance of the gradient of log probability of parameter, i.e., $\text{Cov}_{q_\phi}[\nabla_\phi\log q_\phi(\boldsymbol{x})]$. For $q(\boldsymbol{x}) = \mathcal{N}(\boldsymbol{x}; \boldsymbol{\mu}, \boldsymbol{\Sigma}) = \mathcal{N}(\boldsymbol{x}; \boldsymbol{\mu}, \Lambda^{-1}\boldsymbol{I})$, the Fisher information matrices of mean and precision are given respectively by $\boldsymbol{F}_{\boldsymbol{\mu}} = \Lambda\boldsymbol{I}$ and $\boldsymbol{F}_\Lambda = \frac{1}{2}\Lambda^{-2}\boldsymbol{I}$. Thus, the natural gradients of parameters $\{\boldsymbol{\mu}, \Lambda\}$ have concise forms given by

$$\tilde{\nabla}_{\boldsymbol{\mu}}\mathcal{L} = \boldsymbol{F}_{\boldsymbol{\mu}}^{-1}\nabla_{\boldsymbol{\mu}}\mathbb{E}_q[h(\boldsymbol{x})] = \Lambda^{-1}\mathbb{E}_q[\nabla_{\boldsymbol{x}}h(\boldsymbol{x})]$$
$$\tilde{\nabla}_\Lambda\mathcal{L} = \boldsymbol{F}_\Lambda^{-1}\nabla_\Lambda\mathbb{E}_q[h(\boldsymbol{x})] = -\mathbb{E}_q[\text{Tr}(\nabla_{\boldsymbol{x}}^2 h(\boldsymbol{x}))].$$

(16)

Following the natural gradient given in (16), we update the variational parameters $\phi_k = \{\boldsymbol{\mu}_k, \Lambda_k\}$ of loss function in (14) using natural gradient descent (NGD) as

$$\boldsymbol{\mu}_k \leftarrow \boldsymbol{\mu}_k - s_1\Lambda_k^{-1}\nabla_{\boldsymbol{\mu}_k}\text{KL}(q_k||p_k) = \boldsymbol{\mu}_k + s_1\Lambda_k^{-1}\mathbb{E}_{q_k}[\nabla_{\boldsymbol{x}_k}\log p_k(\boldsymbol{x}_k|\boldsymbol{x}_{k+1}, \boldsymbol{y})]$$
$$\Lambda_k \leftarrow \Lambda_k - 2s_2\Lambda_k^2\nabla_{\Lambda_k}\text{KL}(q_k||p_k) = \Lambda_k - s_2(N\Lambda_k + \mathbb{E}_{q_k}[\text{Tr}(\nabla_{\boldsymbol{x}_k}^2\log p_k(\boldsymbol{x}_k|\boldsymbol{x}_{k+1}, \boldsymbol{y}))])$$

(17)

where $s_1$ and $s_2$ are step sizes. We obtain a stochastic NGD update when the expectations of gradient and Hessian matrix in (17) are approximated by sample mean:

$$\boldsymbol{\mu}_k^{(i+1)} = \boldsymbol{\mu}_k^{(i)} + s_1(\Lambda_k^{(i)})^{-1}\frac{1}{L}\sum_{i=1}^{L}\nabla_{\boldsymbol{x}_k}\log p_k(\boldsymbol{x}_k|\boldsymbol{x}_{k+1}, \boldsymbol{y})|_{\boldsymbol{x}_k = \boldsymbol{x}_k^{(i)} \sim q_k^{(i)}}$$

$$\Lambda_k^{(i+1)} = \Lambda_k^{(i)} - s_2\left(N\Lambda_k^{(i)} + \frac{1}{L}\sum_{i=1}^{L}\text{Tr}(\nabla_{\boldsymbol{x}_k}^2\log p_k(\boldsymbol{x}_k|\boldsymbol{x}_{k+1}, \boldsymbol{y}))|_{\boldsymbol{x}_k = \boldsymbol{x}_k^{(i)} \sim q_k^{(i)}}\right)$$

(18)

where $L$ is the number of samples. The stochastic update of parameters of $q_k$ converges to the local minima of KL divergence $\text{KL}(q_k||p_k)$ which is a Gaussian approximation of the posterior $p_k(\boldsymbol{x}_k|\boldsymbol{x}_{k+1}, \boldsymbol{y})$. It is worth noting that we choose stochastic NGD since it achieves a good performance and parameters involved are easy to tune in our experiments. Other optimization methods may be applied. In stochastic NGD update (18), the gradient and Hessian of $\log p_k(\boldsymbol{x}_k|\boldsymbol{x}_{k+1}, \boldsymbol{y})$ are required. We next introduce some approximations to simplify the calculation.

### 4.3 SCORE-BASED GRADIENT CALCULATION

From Bayes' rule, the score of reverse conditional $p(\boldsymbol{x}_k|\boldsymbol{x}_{k+1}, \boldsymbol{y}) = \frac{p(\boldsymbol{x}_{k+1}|\boldsymbol{x}_k)p(\boldsymbol{x}_k|\boldsymbol{y})}{p(\boldsymbol{x}_{k+1}|\boldsymbol{y})}$ involved in (18) is given by $\nabla_{\boldsymbol{x}_k}\log p_k(\boldsymbol{x}_k|\boldsymbol{x}_{k+1}, \boldsymbol{y}) = \nabla_{\boldsymbol{x}_k}\log p(\boldsymbol{x}_{k+1}|\boldsymbol{x}_k) + \nabla_{\boldsymbol{x}_k}\log p(\boldsymbol{y}|\boldsymbol{x}_k) + \nabla_{\boldsymbol{x}_k}\log p(\boldsymbol{x}_k)$, where $\nabla_{\boldsymbol{x}_k}\log p(\boldsymbol{x}_k)$ is the noisy score function which can be approximated by a well-trained score network $\boldsymbol{s}_{\boldsymbol{\theta}}(\boldsymbol{x}_k, \sigma_k)$ and $\nabla_{\boldsymbol{x}_k}\log p(\boldsymbol{x}_{k+1}|\boldsymbol{x}_k)$ can be calculated explicitly for both VE and VP diffusion models. For VE diffusion $\nabla_{\boldsymbol{x}_k}\log p(\boldsymbol{x}_{k+1}|\boldsymbol{x}_k) = \frac{\boldsymbol{x}_{k+1} - \boldsymbol{x}_k}{\sigma_{k+1}^2 - \sigma_k^2}$, and for VP diffusion $\nabla_{\boldsymbol{x}_k}\log p(\boldsymbol{x}_{k+1}|\boldsymbol{x}_k) = \frac{\sqrt{1-\beta_{k+1}}}{\beta_{k+1}}\boldsymbol{x}_{k+1} - \frac{1-\beta_{k+1}}{\beta_{k+1}}\boldsymbol{x}_k$. However, the likelihood score, i.e.,

the gradient of logarithm conditional $\nabla_{\boldsymbol{x}_k} \log p(\boldsymbol{y}|\boldsymbol{x}_k)$ is hard to handle in general. For linear inverse problems, several SVD based approximations of $\nabla_{\boldsymbol{x}_k} \log p(\boldsymbol{y}|\boldsymbol{x}_k)$ are proposed in Kawar et al. (2021; 2022); Meng & Kabashima (2022) for linear measurements and Gaussian approximation for general measurements are discussed in Song et al. (2023). In Chung et al. (2022a), the authors propose the following approximation that can be applied for general measurements:

$$\log p(\boldsymbol{y}|\boldsymbol{x}_k) \approx \log p(\boldsymbol{y}|\hat{\boldsymbol{x}}_0(\boldsymbol{x}_k)) \tag{19}$$

where $\hat{\boldsymbol{x}}_0(\boldsymbol{x}_k)$ is the MMSE estimate of $\boldsymbol{x}_0$. For VE diffusion, according to the Tweedie formula:

$$\hat{\boldsymbol{x}}_0(\boldsymbol{x}_k) = \mathbb{E}_{p(\boldsymbol{x}_0|\boldsymbol{x}_k)}[\boldsymbol{x}_0] = \boldsymbol{x}_k + \sigma_k^2 \nabla_{\boldsymbol{x}_k} \log p(\boldsymbol{x}_k). \tag{20}$$

Similarly, for VP diffusion, the MMSE estimation of $\boldsymbol{x}_0$ given $\boldsymbol{x}_k$ is

$$\hat{\boldsymbol{x}}_0(\boldsymbol{x}_k) = \mathbb{E}_{p(\boldsymbol{x}_0|\boldsymbol{x}_k)}[\boldsymbol{x}_0] = \frac{1}{\sqrt{\bar{\alpha}_k}}(\boldsymbol{x}_k + (1-\bar{\alpha}_k)\nabla_{\boldsymbol{x}_k} \log p(\boldsymbol{x}_k)). \tag{21}$$

The approximation error of (19) can be quantified with Jenson's Gap as given in Chung et al. (2022a). We choose the likelihood approximation of (19) in our implementation, other approximation methods can be applied to RMP as discussed in Appendix. As a conclusion, the gradient is calculated as

$$\nabla_{\boldsymbol{x}_k} \log p_k(\boldsymbol{x}_k|\boldsymbol{x}_{k+1}, \boldsymbol{y}) \approx \nabla_{\boldsymbol{x}_k} \log p(\boldsymbol{x}_{k+1}|\boldsymbol{x}_k) + \gamma_k \nabla_{\boldsymbol{x}_k} \log p(\boldsymbol{y}|\hat{\boldsymbol{x}}_0(\boldsymbol{x}_k)) + \boldsymbol{s}_{\boldsymbol{\theta}}(\boldsymbol{x}_k, \sigma_k) \tag{22}$$

where the parameter $\gamma_k$ is added to balance the approximated likelihood score and prior score. We set $\gamma_k = \zeta \frac{\|\boldsymbol{s}_{\boldsymbol{\theta}}(\boldsymbol{x}_k, \sigma_k)\|_2}{\|\nabla \log p(\boldsymbol{y}|\hat{\boldsymbol{x}}_0(\boldsymbol{x}_k))\|_2}$ where $\zeta$ is a hyper parameter to be tuned for different problems. The idea behind the strategy is that we should keep a balance between the data score and the likelihood score.

---

**Algorithm 2:** VE/VP-RMP with Score-based Stochastic NGD

**Input** : $\boldsymbol{y}$, $s_1$, $T$, $T_{in}$, $\boldsymbol{x}_T$, $\boldsymbol{\mu}_{T-1}^0$

**1 for** $k = T-1 : 0$ **do**

**2**     For VE $\Lambda_k^{-1} = \frac{\sigma_k^2(\sigma_{k+1}^2 - \sigma_k^2)}{\sigma_{k+1}^2}$ if $k > T_s$ else $\Lambda_k^{-1} = \sigma_{k+1}^2 - \sigma_k^2$ (for VP $\Lambda_k^{-1} = \beta_{k+1}$)

**3**     **for** $i = 0 : T_{in} - 1$ **do**

**4**        $\boldsymbol{\mu}_k^{(i+1)} = \boldsymbol{\mu}_k^{(i)} + s_1 \Lambda_k^{-1} \left(\nabla_{\boldsymbol{x}_k} \log p(\boldsymbol{x}_{k+1}|\boldsymbol{x}_k) + \gamma_k \nabla_{\boldsymbol{x}_k} \log p(\boldsymbol{y}|\hat{\boldsymbol{x}}_0(\boldsymbol{x}_k)) + \boldsymbol{s}_{\boldsymbol{\theta}}(\boldsymbol{x}_k, \sigma_k)\right)$

**5**        where $\boldsymbol{x}_k \sim \mathcal{N}(\boldsymbol{x}; \boldsymbol{\mu}_k^{(i)}, \Lambda_k^{-1}\boldsymbol{I})$, $\hat{\boldsymbol{x}}_0(\boldsymbol{x}_k) = \boldsymbol{x}_k + \sigma_k^2 \boldsymbol{s}_{\boldsymbol{\theta}}(\boldsymbol{x}_k, \sigma_k)$ for VE

**6**        (for VP $\hat{\boldsymbol{x}}_0(\boldsymbol{x}_k) = \frac{1}{\sqrt{\bar{\alpha}_k}}(\boldsymbol{x}_k + (1-\bar{\alpha}_k)\boldsymbol{s}_{\boldsymbol{\theta}}(\boldsymbol{x}_k, \sigma_k))$

**7**     **end**

**8**     $\boldsymbol{x}_k = \boldsymbol{\mu}_k^{(T_{in})}$ and $\boldsymbol{\mu}_{k-1}^{(0)} = \boldsymbol{\mu}_k^{(T_{in})}$

**9 end**

**Output** : $\boldsymbol{\mu}_0^{(T_{in})}$

---

### 4.4 Fixed Precision Update

In the update (18), the Hessian matrix $\nabla_{\boldsymbol{x}_k}^2 \log p_k(\boldsymbol{x}_k|\boldsymbol{x}_{k+1}, \boldsymbol{y})$ is difficult to acquire in general. Also, the complexity involved in the calculation of Hessian may prevent the application of the algorithm. Thus, we introduce an approximation that does not require the calculation of Hessian. According to the Proposition 1, the update of precision $\Lambda_k^{(i+1)}$ converges to the precision of $p_k(\boldsymbol{x}_k|\boldsymbol{x}_{k+1}, \boldsymbol{y})$. Thus, we can fix the update of $\Lambda_k^{(i+1)}$ in Algorithm 1 to the precision of $p_k(\boldsymbol{x}_k|\boldsymbol{x}_{k+1}, \boldsymbol{y})$ and only update $\boldsymbol{\mu}_k$ at each step. According to Proposition 1, for VE diffusion model, if we set $\boldsymbol{C}_{\boldsymbol{x}_0} = v_{\boldsymbol{x}_0}\boldsymbol{I}$, then the inverse of precision is given by $(\Lambda_k^{(i)})^{-1} = \frac{(\sigma_k^2 + v_{\boldsymbol{x}_0})(\sigma_{k+1}^2 - \sigma_k^2)}{\sigma_{k+1}^2 + v_{\boldsymbol{x}_0}}$. We cannot calculate $(\Lambda_k^{(i)})^{-1}$ directly since $v_{\boldsymbol{x}_0}$ is unknown. However, for the cases that $k$ is close to $T$, $\sigma_k^2$ is large enough comparing to $v_{\boldsymbol{x}_0}$. Thus, $(\Lambda_k^{(i)})^{-1}$ can be approximated by $\frac{\sigma_k^2(\sigma_{k+1}^2 - \sigma_k^2)}{\sigma_{k+1}^2}$. For the case that $k$ is close to 0, $\sigma_{k+1}^2, \sigma_k^2 \to 0$, we have $(\Lambda_k^{(i)})^{-1} \approx \sigma_{k+1}^2 - \sigma_k^2$. Similarly, for VP diffusion model, if we set $\boldsymbol{C}_{\boldsymbol{x}_0} = v_{\boldsymbol{x}_0}\boldsymbol{I}$, then $(\Lambda_k^{(i)})^{-1} = \frac{\frac{\beta_{k+1}}{1-\beta_{k+1}}(1-\bar{\alpha}_k + \bar{\alpha}_k v_{\boldsymbol{x}_0})}{\frac{\beta_{k+1}}{1-\beta_{k+1}} + (1-\bar{\alpha}_k + \bar{\alpha}_k v_{\boldsymbol{x}_0})}$. Since $\beta_k \approx 0$ and $\bar{\alpha}_k \approx 0$, we have $(\Lambda_k^{(i)})^{-1} \approx \beta_{k+1}$. With stochastic NGD based VI and score-based approximations of gradient and Hessian, we summarize an algorithm given in Algorithm 2 where $T_{in}$ is the number of NGD steps.

## 5 EXPERIMENTS

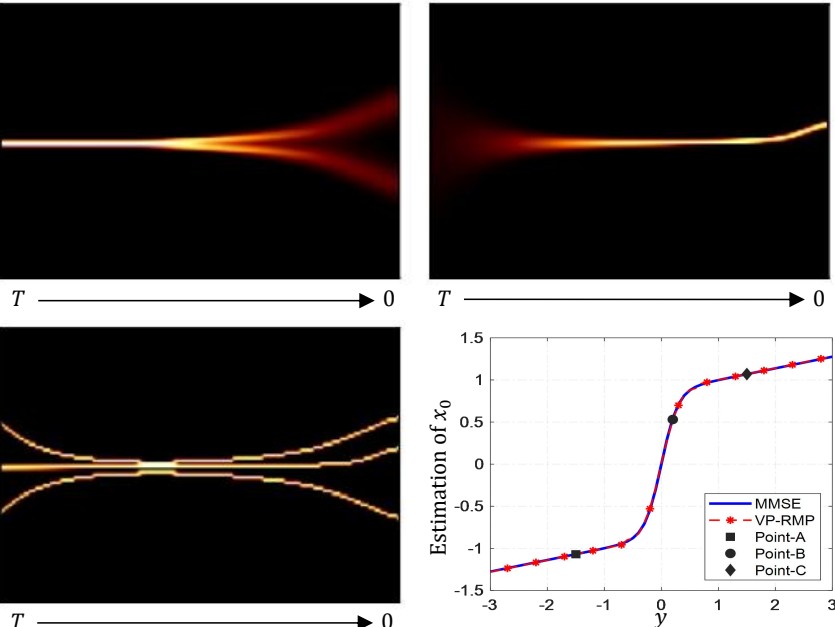

Figure 2: Illustration of the process of VP-based RMP for Gaussian mixture model. Top-left: Various measurement $y$ with fixed $x_T = 0$. Top-right: Random $x_T$ with $y = 0.2$. Bottom-left: $(x_T, y) = (-1, -1.5)$, $(x_T, y) = (0, 0.2)$ and $(x_T, y) = (1, 1.5)$. Bottom-right: MMSE estimation $\mathbb{E}_{p(x_0|y=\bar{y})}[x_0]$ and VP-RMP outputs for different measurement $y$.

### 5.1 GAUSSIAN MIXTURE MODEL

We generate $x_0$ from a Gaussian mixture prior $p(x_0) = \frac{1}{2}\mathcal{N}(x_0; \mu_1, v_1^2) + \frac{1}{2}\mathcal{N}(x_0; \mu_2, v_2^2)$ and get measurement from $y = ax_0 + v_0\varepsilon$, where $\varepsilon \sim \mathcal{N}(0, 1)$, to demonstrate the rationale of RMP. In the experiments, we set $\mu_1 = 1$, $\mu_2 = -1$, $v_1 = v_2 = 0.2$, $v_0 = 0.5$, $a = 1$, and $T = 1000$. Since the model is simple, the data and likelihood score can be calculated exactly (given in Appendix D). By running VP-RMP in Algorithm 2 (with exact score), the propagation of the reverse mean is shown in Figs. 1 and 2. In the right part of Fig. 1, $x_T$ is initialized randomly from $p_T(x_T)$ for various measurement $y$. We see from the plot that the final output of VP-RMP, i.e., the reverse mean $\mu_0$ converges to $\mathbb{E}_{p(x_0|y)}[x_0]$. In the top-left plot of Fig. 2, for various measurement $y$, we initialize $x_T = 0$. The final output of VP-RMP is almost the same as that in Fig. 1 which converges to $\mathbb{E}_{p(x_0|y)}[x_0]$. In the top-right plot of Fig. 2, $x_T$ is initialized randomly from $p_T(x_T)$ and $y$ is set to 0.2. The reverse mean converges to the posterior mean $\mathbb{E}_{p(x_0|y=0.2)}[x_0]$ no matter what $x_T$ is. In the bottom-left plot of Fig. 2, $x_T$ and $y$ are fixed to three cases. We see that the evolution of reverse mean is almost deterministic when the initial point $x_T$ and $y$ are fixed. In the bottom-right plot of Fig. 2, the theoretical results of $\mathbb{E}_{p(x_0|y=\bar{y})}[x_0]$ and estimations of $x_0$ using VP-RMP for various $y$ are shown where points A, B, and C correspond to the outputs of VP-RMP for the three cases given in the bottom-left respectively. We see that the outputs of VP-RMP consists with the posterior mean well.

### 5.2 IMAGE RECONSTRUCTION

We compare our proposed VE/VP-RMP in Algorithm 2 with baseline algorithms for different image reconstruction tasks on FFHQ $256 \times 256$ dataset (Karras et al., 2019). For VP-based methods, the pre-trained score network for FFHQ $256 \times 256$ is take from Chung et al. (2022a) and for VE-based methods, the pre-trained score networks for FFHQ $256 \times 256$ is taken from Song et al. (2020b). The algorithms we compare include Diffusion Posterior Sampling (DPS) (Chung et al., 2022a), Manifold Constrained Gradients (MCG) and Plug and Play ADMM (PnP-ADMM) (Chan

Table 1: Quantitative results (PSNR, SSIM, FID, LPIPS) of solving linear inverse problems with Gaussian noise ($\epsilon = 0.05$) on FFHQ validation dataset. **Bold**: best, Underline: second best.

| Methods | SR (4 ×) | | Inpaint (box) | | Inpaint (random) | | Deblur (Gauss) | | Deblur (motion) | |
|---|---|---|---|---|---|---|---|---|---|---|
| | PSNR ↑ | SSIM ↑ | PSNR ↑ | SSIM ↑ | PSNR ↑ | SSIM ↑ | PSNR ↑ | SSIM ↑ | PSNR ↑ | SSIM ↑ |
| VE-RMP (Ours) | **29.26** | **0.8421** | 25.23 | 0.8361 | **34.58** | 0.9315 | **28.28** | **0.8214** | 27.74 | **0.8232** |
| VP-RMP (Ours) | 28.89 | 0.8410 | **25.78** | **0.8716** | 34.27 | **0.9291** | 27.96 | 0.8153 | **28.10** | 0.7921 |
| DPS | 24.32 | 0.7016 | 24.68 | 0.8182 | 29.85 | 0.8554 | 25.50 | 0.7210 | 23.42 | 0.6712 |
| MCG | 18.16 | 0.2145 | 11.30 | 0.2450 | 11.06 | 0.0812 | 11.65 | 0.1317 | 11.53 | 0.1065 |
| PnP-ADMM | 23.21 | 0.6744 | 13.06 | 0.4534 | 18.49 | 0.5154 | 27.13 | 0.7763 | 24.12 | 0.7812 |
| Methods | SR (4 ×) | | Inpaint (box) | | Inpaint (random) | | Deblur(Gauss) | | Deblur (motion) | |
| | FID ↓ | LPIPS ↓ | FID ↓ | LPIPS ↓ | FID ↓ | LPIPS ↓ | FID ↓ | LPIPS ↓ | FID ↓ | LPIPS ↓ |
| VE-RMP (Ours) | 89.04 | 0.2278 | 96.36 | 0.2372 | 36.37 | 0.0948 | 94.43 | 0.2515 | 90.63 | 0.2438 |
| VP-RMP (Ours) | **57.27** | **0.1890** | **28.36** | **0.1250** | **22.41** | **0.0771** | **59.10** | **0.2024** | **62.80** | **0.2383** |
| DPS | 72.44 | 0.2484 | 53.35 | 0.1905 | 57.74 | 0.1887 | 60.18 | 0.2377 | 68.83 | 0.2576 |
| MCG | 227.65 | 0.6232 | 443.24 | 0.7929 | 486.09 | 0.8224 | 354.01 | 0.7760 | 461.38 | 0.7222 |
| PnP-ADMM | 86.32 | 0.4723 | 165.31 | 0.5099 | 117.51 | 0.4628 | 125.63 | 0.3180 | 182.41 | 0.4212 |

Table 2: Quantitative evaluation (PSNR, SSIM, FID, LPIPS) of solving nonlinear inverse problems on FFHQ validation dataset. **Bold**: best, Underline: second best.

| Methods | Phase retrieval | | | | Nonlinear deblur | | | |
|---|---|---|---|---|---|---|---|---|
| | PSNR | SSIM | FID | LPIPS | PSNR | SSIM | FID | LPIPS |
| VE-RMP (Ours) | **27.98** | **0.7860** | 72.84 | 0.1857 | **25.21** | **0.7222** | 143.02 | 0.3401 |
| VP-RMP (Ours) | 25.32 | 0.7632 | **62.11** | **0.1512** | 24.33 | 0.6872 | **66.47** | **0.2219** |
| DPS | 12.75 | 0.4142 | 218.78 | 0.5828 | 23.62 | 0.6696 | 76.85 | 0.2685 |

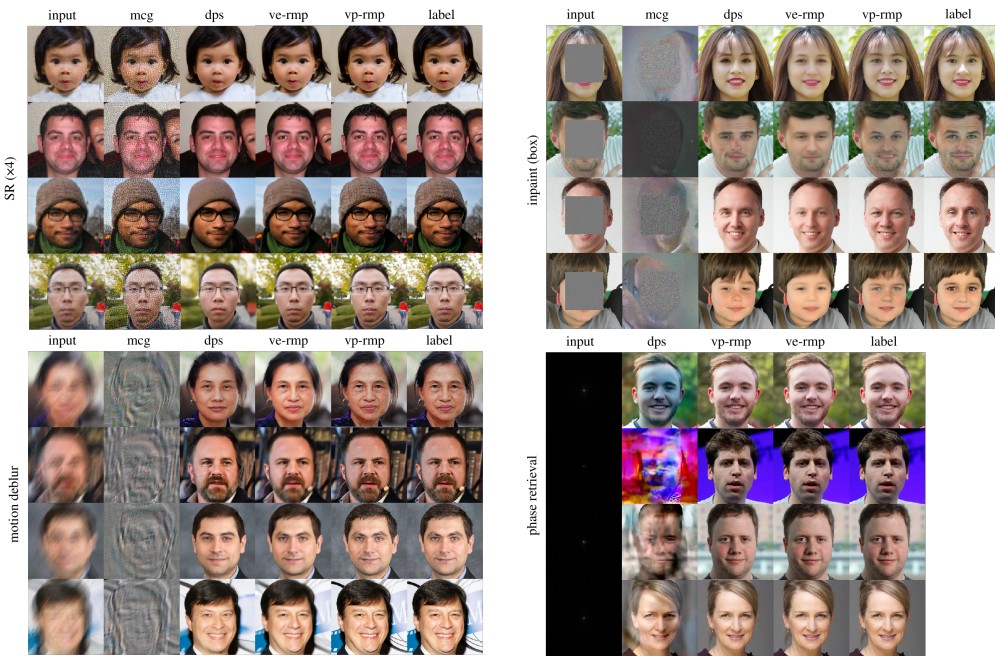

Figure 3: Part of the results on solving inverse problems with Gaussian noise ($\epsilon = 0.05$).

et al., 2016). Denoising Diffusion Restoration Models (DDRM) and ΠGDM (Song et al., 2023) are discussed in the appendix part since these methods have difference likelihood score approximation and DDRM can only be applied for linear measurements. Unless otherwise specified, the measurement noise is set to Gaussian. The four metrics are used for comparisons: Frechet Inception Distance (FID) (Heusel et al., 2017), Learned Perceptual Image Patch Similarity (LPIPS) (Zhang et al., 2018), Peak Signal-to-Noise-Ratio (PSNR) and Structural Similarity Index Measure (SSIM) (Wang et al., 2004).

For VP-based algorithms i.e., DPS, MCG and VP-RMP, $\beta_k$ varies linearly with $k$ in range from 0.0001 to 0.02 as given in Ho et al. (2020). The timesteps for DPS, MCG, and VP-RMP are all fixed to $T = 400$. The inner loop size $T_{in} = 1$ for VP-RMP. For VE-RMP, we set $T = 30$ and $T_{in} = 20$ for all tasks. $\{\sigma_k\}, k = 1, 2 \cdots, T$ is set as a positive geometric sequence as suggested in Song & Ermon (2019) that satisfies $\frac{\sigma_k}{\sigma_{k+1}} > 1$. In all experiments, we set $\sigma_k = \sigma_{min}\left(\frac{\sigma_{max}}{\sigma_{min}}\right)^{\frac{k-1}{T-1}}$ where $\sigma_{min} = 0.01$ and $\sigma_{max} = 100$ that matches the pre-trained score network given in Song et al. (2020b). The settings of step size $s_1$ and $\zeta$ for VE/VP-RMP are given in Appendix E. Other parameters in DPS, MCG are set according to the configurations provided by its' authors. The code of our algorithms is developed under the framework developed by DPS, and the measurement operator used in our comparisons are the same for all algorithms such that the comparisons are fair enough for all algorithms. Our code is available at `https://github.com/neuripsrmp/rmp.git`.

**Linear Inverse Problems:** We compare algorithms on image reconstruction problems including super resolution, inpainting (box, random) and deblur (Gaussian, motion deblur). The evaluation results on different metrics are given in Tables 1. We see from Tables 1 that VE/VP-RMP outperforms other algorithms on almost all tasks and VP-RMP achieves the best performance in both FID and LPIPS metrics in all tasks. We present some results in Fig. 3. More results on FFHQ and ImageNet (Russakovsky et al., 2015) datasets are given in Appendix F.

**Nonlinear Inverse Problems:** We compare VE/VP-based RMP with DPS on non-linear image reconstruction tasks including phase retrieval and non-linear deblur on FFHQ dataset. The comparison results are presented in Table 2. We see that similarly to the linear cases, RMP achieves better performance comparing to DPS in most of the metrics.

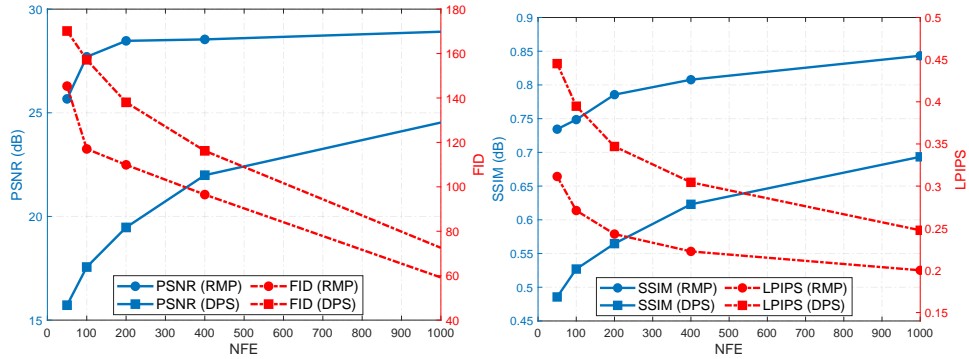

Figure 4: Performance comparisons of algorithms with different NFE on SR task.

## 5.3 Complexity vs Performance

The complexity of diffusion-based methods dependent on the number of neural function evaluations (NFE). RMP consists of $T$ outer loops and each outer loop has $T_{in}$ inner loops. The operations in the outer loops can be ignored since only variable assignments and scalar calculation are involved. In each inner loop, two main operations are involved: denoising with score network $s_\theta(x_k, \sigma_k)$ and stochastic natural gradient descent step which requires the likelihood score. Thus, the NFE of VE/VP-RMP is $TT_{in}$. To show the performance of RMPs with different NFE, we compare the curves of NFE versus the performance of VP-RMP and DPS on SR task in Fig. 4. From the figure, we see that RMP outperforms DPS significantly on all metrics for all NFEs. We also note that for the PSNR metric, VP-RMP with 50 NFEs achieves a higher PSNR than DPS with 1000 NFEs.

## 6 Conclusions

In this paper, we propose a score-based variational inference framework for inverse problems. By optimizing the reverse KL divergence at each step of the reverse process and tracking the evolution of reverse mean at each reverse step, a practical algorithm for general inverse problems is proposed. Extensive experiments validate our theoretical results and demonstrate that our proposed algorithm achieves superior performance on various image reconstruction tasks.

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

## A PROOF OF PROPOSITION 1

### A.1 GAUSSIAN PROPERTY

First, the reverse conditional $p_k(\boldsymbol{x}_k|\boldsymbol{x}_{k+1},\boldsymbol{y})$ is given by

$$p_k(\boldsymbol{x}_k|\boldsymbol{x}_{k+1},\boldsymbol{y}) = \frac{p(\boldsymbol{x}_{k+1}|\boldsymbol{x}_k)p(\boldsymbol{x}_k|\boldsymbol{y})}{p(\boldsymbol{x}_{k+1}|\boldsymbol{y})} \tag{23}$$

where $\boldsymbol{x}_k$ and $\boldsymbol{x}_{k+1}$ can be viewed as the discretization of continuous diffusion process of $\boldsymbol{x}_t$. We relate $\{\boldsymbol{x}_k\}_{k=0}^T$ to continuous stochastic process $\{\boldsymbol{x}_t\}_{t=0}^1$ by letting $\boldsymbol{x}_k = \boldsymbol{x}_{t=k\Delta t}$, where $\Delta t = \frac{1}{T}$. The discrete $\{\boldsymbol{x}_k\}_{k=0}^T$ becomes a continuous process in the limit $\Delta t \to 0$. Thus, we focus on the continuous form of the reverse conditional, i.e., $p_k(\boldsymbol{x}_k|\boldsymbol{x}_{k+1}, \boldsymbol{y}) = p(\boldsymbol{x}_t|\boldsymbol{x}_{t+\Delta t}, \boldsymbol{y})$. The reverse conditional in a continuous form is given by

$$p(\boldsymbol{x}_t|\boldsymbol{x}_{t+\Delta t}, \boldsymbol{y}) = \frac{p(\boldsymbol{x}_{t+\Delta t}|\boldsymbol{x}_t)p(\boldsymbol{x}_t|\boldsymbol{y})}{p(\boldsymbol{x}_{t+\Delta t}|\boldsymbol{y})}. \tag{24}$$

As shown in Song et al. (2020b), the forward process can be characterized by the following SDE:

$$d\boldsymbol{x} = \boldsymbol{f}_t(\boldsymbol{x})dt + g_t(\boldsymbol{x})d\bar{\boldsymbol{w}} \tag{25}$$

where $\bar{\boldsymbol{w}}$ is the standard Wiener process. For VE diffusion, $\boldsymbol{f}_t(\boldsymbol{x}) = 0$ and $g_t(\boldsymbol{x}) = \sqrt{\frac{d\sigma(t)^2}{dt}}$, and for VP diffusion, $\boldsymbol{f}_t(\boldsymbol{x}) = -\frac{1}{2}\beta(t)\boldsymbol{x}$ and $g_t(\boldsymbol{x}) = \sqrt{\beta(t)}$. When $\Delta t \to 0$ we have

$$\boldsymbol{x}_{t+\Delta t} - \boldsymbol{x}_t = \boldsymbol{f}_t(\boldsymbol{x}_t)\Delta t + g_t(\boldsymbol{x}_t)\sqrt{\Delta t}\boldsymbol{\varepsilon} \tag{26}$$

where $\boldsymbol{\varepsilon} \sim \mathcal{N}(0, \boldsymbol{I})$. Then

$$p(\boldsymbol{x}_{t+\Delta t}|\boldsymbol{x}_t) = \mathcal{N}(\boldsymbol{x}_{t+\Delta t}; \boldsymbol{x}_t + \boldsymbol{f}_t(\boldsymbol{x}_t)\Delta t, g_t^2(\boldsymbol{x}_t)\Delta t)$$
$$\propto \exp\left(-\frac{\|\boldsymbol{x}_{t+\Delta t} - \boldsymbol{x}_t - \boldsymbol{f}_t(\boldsymbol{x}_t)\Delta t\|_2^2}{2g_t^2(\boldsymbol{x}_t)\Delta t}\right) \tag{27}$$

and

$$p(\boldsymbol{x}_t|\boldsymbol{x}_{t+\Delta t}, \boldsymbol{y}) \propto \exp\left(-\frac{\|\boldsymbol{x}_{t+\Delta t} - \boldsymbol{x}_t - \boldsymbol{f}_t(\boldsymbol{x}_t)\Delta t\|_2^2}{2g_t^2(\boldsymbol{x}_t)\Delta t} + \log p(\boldsymbol{x}_t|\boldsymbol{y}) - \log p(\boldsymbol{x}_{t+\Delta t}|\boldsymbol{y})\right). \tag{28}$$

When $\Delta t \to 0$, $\log p(\boldsymbol{x}_t|\boldsymbol{y})$ can be expressed by Taylor expansion:

$$\log p(\boldsymbol{x}_{t+\Delta t}|\boldsymbol{y}) \approx \log p(\boldsymbol{x}_t|\boldsymbol{y}) + (\boldsymbol{x}_{t+\Delta t} - \boldsymbol{x}_t)^T \nabla_{\boldsymbol{x}_t} \log p(\boldsymbol{x}_t|\boldsymbol{y}) + \Delta t \frac{\partial}{\partial t} \log p(\boldsymbol{x}_t|\boldsymbol{y}). \tag{29}$$

Thus, we have

$$p(\boldsymbol{x}_t|\boldsymbol{x}_{t+\Delta t}, \boldsymbol{y}) \propto \exp\left(-\frac{\|\boldsymbol{x}_{t+\Delta t} - \boldsymbol{x}_t - (\boldsymbol{f}_t(\boldsymbol{x}_t) - g_t^2(\boldsymbol{x}_t)\nabla_{\boldsymbol{x}_t} \log p(\boldsymbol{x}_t|\boldsymbol{y}))\Delta t\|_2^2}{2g_t^2(\boldsymbol{x}_t)\Delta t} + \mathcal{O}(\Delta t)\right) \tag{30}$$

which is Gaussian when $\Delta t \to 0$. Thus, $p_k(\boldsymbol{x}_k|\boldsymbol{x}_{k+1}, \boldsymbol{y})$ is Gaussian when $\Delta t \to 0$.

## A.2 VE DIFFUSION

For VE diffusion models, the mean of $p(\boldsymbol{x}_k|\boldsymbol{y})$ is given by

$$\begin{aligned}
\mathbb{E}_{p(\boldsymbol{x}_k|\boldsymbol{y})}[\boldsymbol{x}_k] &= \int \boldsymbol{x}_k p(\boldsymbol{x}_k|\boldsymbol{y})d\boldsymbol{x}_k \\
&= \int \boldsymbol{x}_k \int p(\boldsymbol{x}_k|\boldsymbol{x}_0)p(\boldsymbol{x}_0|\boldsymbol{y})d\boldsymbol{x}_0 d\boldsymbol{x}_k \\
&= \int \int \boldsymbol{x}_k p(\boldsymbol{x}_k|\boldsymbol{x}_0)d\boldsymbol{x}_k p(\boldsymbol{x}_0|\boldsymbol{y})d\boldsymbol{x}_0 \\
&= \int \boldsymbol{x}_0 p(\boldsymbol{x}_0|\boldsymbol{y})d\boldsymbol{x}_0 = \mathbb{E}_{p(\boldsymbol{x}_0|\boldsymbol{y})}[\boldsymbol{x}_0]
\end{aligned} \tag{31}$$

and its covariance matrix is given by

$$
\begin{aligned}
&\mathrm{Cov}_{p(\boldsymbol{x}_k|\boldsymbol{y})}[\boldsymbol{x}_k] \\
&= \int \boldsymbol{x}_k \boldsymbol{x}_k^T p(\boldsymbol{x}_k|\boldsymbol{y})d\boldsymbol{x}_k - \mathbb{E}_{p(\boldsymbol{x}_0|\boldsymbol{y})}[\boldsymbol{x}_0]\mathbb{E}_{p(\boldsymbol{x}_0|\boldsymbol{y})}[\boldsymbol{x}_0]^T \\
&= \int \boldsymbol{x}_k \boldsymbol{x}_k^T \int p(\boldsymbol{x}_k|\boldsymbol{x}_0)p(\boldsymbol{x}_0|\boldsymbol{y})d\boldsymbol{x}_0 d\boldsymbol{x}_k - \mathbb{E}_{p(\boldsymbol{x}_0|\boldsymbol{y})}[\boldsymbol{x}_0]\mathbb{E}_{p(\boldsymbol{x}_0|\boldsymbol{y})}[\boldsymbol{x}_0]^T \\
&= \int \int \boldsymbol{x}_k \boldsymbol{x}_k^T p(\boldsymbol{x}_k|\boldsymbol{x}_0)d\boldsymbol{x}_k p(\boldsymbol{x}_0|\boldsymbol{y})d\boldsymbol{x}_0 - \mathbb{E}_{p(\boldsymbol{x}_0|\boldsymbol{y})}[\boldsymbol{x}_0]\mathbb{E}_{p(\boldsymbol{x}_0|\boldsymbol{y})}[\boldsymbol{x}_0]^T \\
&= \int (\boldsymbol{x}_0\boldsymbol{x}_0^T + \sigma_k^2\boldsymbol{I})p(\boldsymbol{x}_0|\boldsymbol{y})d\boldsymbol{x}_0 - \mathbb{E}_{p(\boldsymbol{x}_0|\boldsymbol{y})}[\boldsymbol{x}_0]\mathbb{E}_{p(\boldsymbol{x}_0|\boldsymbol{y})}[\boldsymbol{x}_0]^T \\
&= \sigma_k^2\boldsymbol{I} + \mathrm{Cov}_{p(\boldsymbol{x}_0|\boldsymbol{y})}[\boldsymbol{x}_0] + \mathbb{E}_{p(\boldsymbol{x}_0|\boldsymbol{y})}[\boldsymbol{x}_0]\mathbb{E}_{p(\boldsymbol{x}_0|\boldsymbol{y})}[\boldsymbol{x}_0]^T - \mathbb{E}_{p(\boldsymbol{x}_0|\boldsymbol{y})}[\boldsymbol{x}_0]\mathbb{E}_{p(\boldsymbol{x}_0|\boldsymbol{y})}[\boldsymbol{x}_0]^T \\
&= \sigma_k^2\boldsymbol{I} + \mathrm{Cov}_{p(\boldsymbol{x}_0|\boldsymbol{y})}[\boldsymbol{x}_0] = \sigma_k^2\boldsymbol{I} + \boldsymbol{C}_{\boldsymbol{x}_0}.
\end{aligned}
\tag{32}
$$

Since $p_k(\boldsymbol{x}_k|\boldsymbol{x}_{k+1}, \boldsymbol{y})$ can be expanded as a Gaussian around $\boldsymbol{x}_{k+1}$ using the second order Taylor approximation when $\Delta t \to 0$ and $p(\boldsymbol{x}_{k+1}|\boldsymbol{x}_k) = \mathcal{N}(\boldsymbol{x}_{k+1}; \boldsymbol{x}_k, (\sigma_{k+1}^2 - \sigma_k^2)\boldsymbol{I})$, then $p(\boldsymbol{x}_k|\boldsymbol{y})$ can also be expanded around $\boldsymbol{x}_{k+1}$ and approximately expressed in the form of a Gaussian when $\Delta t \to 0$, i.e., $p(\boldsymbol{x}_k|\boldsymbol{y}) \approx \mathcal{N}(\boldsymbol{x}_k; \mathbb{E}_{p(\boldsymbol{x}_0|\boldsymbol{y})}[\boldsymbol{x}_0], (\sigma_k^2\boldsymbol{I} + \boldsymbol{C}_{\boldsymbol{x}_0}))$. For two multivariate Gaussian distribution $G_1(\boldsymbol{x}) = \mathcal{N}(\boldsymbol{x}, \boldsymbol{\mu}_1, \boldsymbol{\Sigma}_1)$ and $G_2(\boldsymbol{x}) = \mathcal{N}(\boldsymbol{x}, \boldsymbol{\mu}_2, \boldsymbol{\Sigma}_2)$, the product $G_1(\boldsymbol{x})G_2(\boldsymbol{x})$ is also Gaussian with mean and covariance given by

$$
\begin{aligned}
\boldsymbol{\mu}_3 &= \boldsymbol{\Sigma}_2(\boldsymbol{\Sigma}_1 + \boldsymbol{\Sigma}_2)^{-1}\boldsymbol{\mu}_1 + \boldsymbol{\Sigma}_1(\boldsymbol{\Sigma}_1 + \boldsymbol{\Sigma}_2)^{-1}\boldsymbol{\mu}_2 \\
\boldsymbol{\Sigma}_3 &= \boldsymbol{\Sigma}_1(\boldsymbol{\Sigma}_1 + \boldsymbol{\Sigma}_2)^{-1}\boldsymbol{\Sigma}_2.
\end{aligned}
\tag{33}
$$

Thus, the mean and covariance of $p_k(\boldsymbol{x}_k|\boldsymbol{x}_{k+1}, \boldsymbol{y})$ can be calculated according to (33) and are given respectively by:

$$
\begin{aligned}
\boldsymbol{\mu}_k &= (\sigma_k^2\boldsymbol{I} + \boldsymbol{C}_{\boldsymbol{x}_0})(\sigma_{k+1}^2\boldsymbol{I} + \boldsymbol{C}_{\boldsymbol{x}_0})^{-1}\boldsymbol{x}_{k+1} + (\sigma_{k+1}^2 - \sigma_k^2)(\sigma_{k+1}^2\boldsymbol{I} + \boldsymbol{C}_{\boldsymbol{x}_0})^{-1}\mathbb{E}_{p(\boldsymbol{x}_0|\boldsymbol{y})}[\boldsymbol{x}_0] \\
&= \boldsymbol{V}_{k,p_1}\boldsymbol{x}_{k+1} + \boldsymbol{V}_{k,p_2}\mathbb{E}_{p(\boldsymbol{x}_0|\boldsymbol{y})}[\boldsymbol{x}_0] \\
\boldsymbol{C}_k &= (\sigma_{k+1}^2 - \sigma_k^2)(\sigma_k^2\boldsymbol{I} + \boldsymbol{C}_{\boldsymbol{x}_0})(\sigma_{k+1}^2\boldsymbol{I} + \boldsymbol{C}_{\boldsymbol{x}_0})^{-1}.
\end{aligned}
\tag{34}
$$

## A.3 VP DIFFUSION

For VP diffusion models, the mean of $p(\boldsymbol{x}_k|\boldsymbol{y})$ is given by

$$
\begin{aligned}
\mathbb{E}_{p(\boldsymbol{x}_k|\boldsymbol{y})}[\boldsymbol{x}_k] &= \int \boldsymbol{x}_k p(\boldsymbol{x}_k|\boldsymbol{y})d\boldsymbol{x}_k \\
&= \int \boldsymbol{x}_k \int p(\boldsymbol{x}_k|\boldsymbol{x}_0)p(\boldsymbol{x}_0|\boldsymbol{y})d\boldsymbol{x}_0 d\boldsymbol{x}_k \\
&= \int \int \boldsymbol{x}_k p(\boldsymbol{x}_k|\boldsymbol{x}_0)d\boldsymbol{x}_k p(\boldsymbol{x}_0|\boldsymbol{y})d\boldsymbol{x}_0 \\
&= \int \sqrt{\bar{\alpha}_k}\boldsymbol{x}_0 p(\boldsymbol{x}_0|\boldsymbol{y})d\boldsymbol{x}_0 = \sqrt{\bar{\alpha}_k}\mathbb{E}_{p(\boldsymbol{x}_0|\boldsymbol{y})}[\boldsymbol{x}_0]
\end{aligned}
\tag{35}
$$

and its covariance matrix is given by

$$
\begin{aligned}
\mathrm{Cov}_{p(\boldsymbol{x}_k|\boldsymbol{y})}[\boldsymbol{x}_k] &= \int \boldsymbol{x}_k\boldsymbol{x}_k^T p(\boldsymbol{x}_k|\boldsymbol{y})d\boldsymbol{x}_k - \bar{\alpha}_k\mathbb{E}_{p(\boldsymbol{x}_0|\boldsymbol{y})}[\boldsymbol{x}_0]\mathbb{E}_{p(\boldsymbol{x}_0|\boldsymbol{y})}[\boldsymbol{x}_0]^T \\
&= \int \boldsymbol{x}_k\boldsymbol{x}_k^T \int p(\boldsymbol{x}_k|\boldsymbol{x}_0)p(\boldsymbol{x}_0|\boldsymbol{y})d\boldsymbol{x}_0 d\boldsymbol{x}_k - \bar{\alpha}_k\mathbb{E}_{p(\boldsymbol{x}_0|\boldsymbol{y})}[\boldsymbol{x}_0]\mathbb{E}_{p(\boldsymbol{x}_0|\boldsymbol{y})}[\boldsymbol{x}_0]^T \\
&= \int \int \boldsymbol{x}_k\boldsymbol{x}_k^T p(\boldsymbol{x}_k|\boldsymbol{x}_0)d\boldsymbol{x}_k p(\boldsymbol{x}_0|\boldsymbol{y})d\boldsymbol{x}_0 - \bar{\alpha}_k\mathbb{E}_{p(\boldsymbol{x}_0|\boldsymbol{y})}[\boldsymbol{x}_0]\mathbb{E}_{p(\boldsymbol{x}_0|\boldsymbol{y})}[\boldsymbol{x}_0]^T \\
&= \int (\bar{\alpha}_k\boldsymbol{x}_0\boldsymbol{x}_0^T + (1-\bar{\alpha}_k)\boldsymbol{I})p(\boldsymbol{x}_0|\boldsymbol{y})d\boldsymbol{x}_0 - \bar{\alpha}_k\mathbb{E}_{p(\boldsymbol{x}_0|\boldsymbol{y})}[\boldsymbol{x}_0]\mathbb{E}_{p(\boldsymbol{x}_0|\boldsymbol{y})}[\boldsymbol{x}_0]^T \\
&= (1-\bar{\alpha}_k)\boldsymbol{I} + \bar{\alpha}_k(\mathrm{Cov}_{p(\boldsymbol{x}_0|\boldsymbol{y})}[\boldsymbol{x}_0] + \mathbb{E}_{p(\boldsymbol{x}_0|\boldsymbol{y})}[\boldsymbol{x}_0]\mathbb{E}_{p(\boldsymbol{x}_0|\boldsymbol{y})}[\boldsymbol{x}_0]^T) \\
&\quad - \bar{\alpha}_k\mathbb{E}_{p(\boldsymbol{x}_0|\boldsymbol{y})}[\boldsymbol{x}_0]\mathbb{E}_{p(\boldsymbol{x}_0|\boldsymbol{y})}[\boldsymbol{x}_0]^T \\
&= (1-\bar{\alpha}_k)\boldsymbol{I} + \bar{\alpha}_k\mathrm{Cov}_{p(\boldsymbol{x}_0|\boldsymbol{y})}[\boldsymbol{x}_0] = (1-\bar{\alpha}_k)\boldsymbol{I} + \bar{\alpha}_k\boldsymbol{C}_{\boldsymbol{x}_0}
\end{aligned}
\tag{36}
$$

Similarly to the case of VE diffusion, since $p(\boldsymbol{x}_{k+1}|\boldsymbol{x}_k) = \mathcal{N}(\boldsymbol{x}_{k+1}; \sqrt{1-\beta_{k+1}}\boldsymbol{x}_k, \beta_{k+1}\boldsymbol{I})$, the mean and covariance of $p_k(\boldsymbol{x}_k|\boldsymbol{x}_{k+1}, \boldsymbol{y})$ are given respectively by:

$$
\begin{aligned}
\boldsymbol{\mu}_k &= ((1-\bar{\alpha}_k)\boldsymbol{I} + \bar{\alpha}_k\boldsymbol{C}_{\boldsymbol{x}_0})\left(\left(\frac{\beta_{k+1}}{1-\beta_{k+1}} + 1 - \bar{\alpha}_k\right)\boldsymbol{I} + \bar{\alpha}_k\boldsymbol{C}_{\boldsymbol{x}_0}\right)^{-1}\frac{\boldsymbol{x}_{k+1}}{\sqrt{1-\beta_{k+1}}} \\
&\quad + \frac{\beta_{k+1}}{1-\beta_{k+1}}\left(\left(\frac{\beta_{k+1}}{1-\beta_{k+1}} + 1 - \bar{\alpha}_k\right)\boldsymbol{I} + \bar{\alpha}_k\boldsymbol{C}_{\boldsymbol{x}_0}\right)^{-1}\sqrt{\bar{\alpha}_k}\mathbb{E}_{p(\boldsymbol{x}_0|\boldsymbol{y})}[\boldsymbol{x}_0] \\
&= ((1-\bar{\alpha}_k)\boldsymbol{I} + \bar{\alpha}_k\boldsymbol{C}_{\boldsymbol{x}_0})((1-\bar{\alpha}_{k+1})\boldsymbol{I} + \bar{\alpha}_{k+1}\boldsymbol{C}_{\boldsymbol{x}_0})^{-1}\sqrt{\alpha_{k+1}}\boldsymbol{x}_{k+1} \\
&\quad + (1-\alpha_{k+1})((1-\bar{\alpha}_{k+1})\boldsymbol{I} + \bar{\alpha}_{k+1}\boldsymbol{C}_{\boldsymbol{x}_0})^{-1}\sqrt{\bar{\alpha}_k}\mathbb{E}_{p(\boldsymbol{x}_0|\boldsymbol{y})}[\boldsymbol{x}_0] \\
&= \boldsymbol{V}_{k,p_3}\boldsymbol{x}_{k+1} + \boldsymbol{V}_{k,p_4}\mathbb{E}_{p(\boldsymbol{x}_0|\boldsymbol{y})}[\boldsymbol{x}_0]
\end{aligned}
\tag{37}
$$

and

$$
\boldsymbol{C}_k = \frac{\beta_{k+1}}{1-\beta_{k+1}}((1-\bar{\alpha}_k)\boldsymbol{I} + \bar{\alpha}_k\boldsymbol{C}_{\boldsymbol{x}_0})\left(\left(\frac{\beta_{k+1}}{1-\beta_{k+1}} + 1 - \bar{\alpha}_k\right)\boldsymbol{I} + \bar{\alpha}_k\boldsymbol{C}_{\boldsymbol{x}_0}\right)^{-1}.
\tag{38}
$$

# B  PROOF OF THEOREM 1

## B.1  VE DIFFUSION

For VE diffusion model, if we set $\boldsymbol{x}_k = \boldsymbol{\mu}_k(\boldsymbol{x}_{k+1} = \boldsymbol{\mu}_{k+1}, \boldsymbol{y}), \forall k = 0 : T-1$, then from Proposition 1, we have

$$
\begin{aligned}
&\boldsymbol{\mu}_0(\boldsymbol{x}_1, \boldsymbol{y}) \\
&= \boldsymbol{V}_{0,p_1}\boldsymbol{x}_1 + \boldsymbol{V}_{0,p_2}\mathbb{E}_{p(\boldsymbol{x}_0|\boldsymbol{y})}[\boldsymbol{x}_0] \\
&= \boldsymbol{V}_{0,p_1}(\boldsymbol{V}_{1,p_1}\boldsymbol{x}_2 + \boldsymbol{V}_{1,p_2}\mathbb{E}_{p(\boldsymbol{x}_0|\boldsymbol{y})}[\boldsymbol{x}_0]) + \boldsymbol{V}_{0,p_2}\mathbb{E}_{p(\boldsymbol{x}_0|\boldsymbol{y})}[\boldsymbol{x}_0] \\
&= \boldsymbol{V}_{0,p_1}\boldsymbol{V}_{1,p_1}\boldsymbol{x}_2 + \boldsymbol{V}_{0,p_1}\boldsymbol{V}_{1,p_2}\mathbb{E}_{p(\boldsymbol{x}_0|\boldsymbol{y})}[\boldsymbol{x}_0] + \boldsymbol{V}_{0,p_2}\mathbb{E}_{p(\boldsymbol{x}_0|\boldsymbol{y})}[\boldsymbol{x}_0] \\
&= \cdots \\
&= \prod_{i=0}^{T-1}\boldsymbol{V}_{i,p_1}\boldsymbol{\mu}_T + (\boldsymbol{V}_{0,p_1}(\boldsymbol{V}_{1,p_1}(\cdots\boldsymbol{V}_{T-2,p_1}(\boldsymbol{V}_{T-1,p_2}) + \boldsymbol{V}_{T-2,p_2}) + \boldsymbol{V}_{1,p_2}) + \boldsymbol{V}_{0,p_2})\mathbb{E}_{p(\boldsymbol{x}_0|\boldsymbol{y})}[\boldsymbol{x}_0]).
\end{aligned}
\tag{39}
$$

For the first part, the coefficient of $\boldsymbol{\mu}_T$ equals

$$
\prod_{i=0}^{T-1}\boldsymbol{V}_{i,p_1} = \prod_{i=0}^{T-1}(\sigma_i^2\boldsymbol{I} + \boldsymbol{C}_{\boldsymbol{x}_0})(\sigma_{i+1}^2\boldsymbol{I} + \boldsymbol{C}_{\boldsymbol{x}_0})^{-1} = (\sigma_0^2\boldsymbol{I} + \boldsymbol{C}_{\boldsymbol{x}_0})(\sigma_T^2\boldsymbol{I} + \boldsymbol{C}_{\boldsymbol{x}_0})^{-1}.
\tag{40}
$$

For the second part, $\forall k = 1 : T-1$, we have

$$
\begin{aligned}
\boldsymbol{V}_{k-1,p_1}\boldsymbol{V}_{k,p_2} + \boldsymbol{V}_{k-1,p_2} &= (\sigma_{k-1}^2\boldsymbol{I} + \boldsymbol{C}_{\boldsymbol{x}_0})(\sigma_k^2\boldsymbol{I} + \boldsymbol{C}_{\boldsymbol{x}_0})^{-1}(\sigma_{k+1}^2 - \sigma_k^2)(\sigma_{k+1}^2\boldsymbol{I} + \boldsymbol{C}_{\boldsymbol{x}_0})^{-1} \\
&\quad + (\sigma_k^2 - \sigma_{k-1}^2)(\sigma_k^2\boldsymbol{I} + \boldsymbol{C}_{\boldsymbol{x}_0})^{-1} \\
&= (\sigma_{k+1}^2 - \sigma_{k-1}^2)(\sigma_{k+1}^2\boldsymbol{I} + \boldsymbol{C}_{\boldsymbol{x}_0})^{-1}.
\end{aligned}
$$

Similarly, we get

$$(\boldsymbol{V}_{0,p_1}(\boldsymbol{V}_{1,p_1}(\cdots \boldsymbol{V}_{T-2,p_1}(\boldsymbol{V}_{T-1,p_2}) + \boldsymbol{V}_{T-2,p_2}) + \boldsymbol{V}_{1,p_2}) + \boldsymbol{V}_{0,p_2}) = (\sigma_T^2 - \sigma_0^2)(\sigma_T^2 \boldsymbol{I} + \boldsymbol{C}_{\boldsymbol{x}_0})^{-1} \tag{41}$$

Thus, we have

$$\boldsymbol{\mu}_0 = (\sigma_0^2 \boldsymbol{I} + \boldsymbol{C}_{\boldsymbol{x}_0})(\sigma_T^2 \boldsymbol{I} + \boldsymbol{C}_{\boldsymbol{x}_0})^{-1} \boldsymbol{\mu}_T + (\sigma_T^2 - \sigma_0^2)(\sigma_T^2 \boldsymbol{I} + \boldsymbol{C}_{\boldsymbol{x}_0})^{-1} \mathbb{E}_{p(\boldsymbol{x}_0|\boldsymbol{y})}[\boldsymbol{x}_0]. \tag{42}$$

When $\sigma_T^2 \to \infty$, $\boldsymbol{\mu}_0(\boldsymbol{x}_1, \boldsymbol{y}) \to \mathbb{E}_{p(\boldsymbol{x}_0|\boldsymbol{y})}[\boldsymbol{x}_0]$, which is the posterior mean.

### B.2  VP Diffusion

Similarly to VE diffusion, for VP diffusion model,

$$\begin{aligned}
&\boldsymbol{\mu}_0(\boldsymbol{x}_1, \boldsymbol{y}) \\
&= \boldsymbol{V}_{0,p_3}\boldsymbol{x}_1 + \boldsymbol{V}_{0,p_4}\mathbb{E}_{p(\boldsymbol{x}_0|\boldsymbol{y})}[\boldsymbol{x}_0] \\
&= \prod_{i=0}^{T-1} \boldsymbol{V}_{i,p_3}\boldsymbol{\mu}_T + (\boldsymbol{V}_{0,p_3}(\boldsymbol{V}_{1,p_3}(\cdots \boldsymbol{V}_{T-2,p_3}(\boldsymbol{V}_{T-1,p_4}) + \boldsymbol{V}_{T-2,p_4}) + \boldsymbol{V}_{1,p_4}) + \boldsymbol{V}_{0,p_4})\mathbb{E}_{p(\boldsymbol{x}_0|\boldsymbol{y})}[\boldsymbol{x}_0])
\end{aligned} \tag{43}$$

For the first part, the coefficient of $\boldsymbol{\mu}_T$ equals

$$\begin{aligned}
\prod_{i=0}^{T-1} \boldsymbol{V}_{i,p_3} &= \prod_{i=0}^{T-1} ((1 - \bar{\alpha}_i)\boldsymbol{I} + \bar{\alpha}_i \boldsymbol{C}_{\boldsymbol{x}_0})((1 - \bar{\alpha}_{i+1})\boldsymbol{I} + \bar{\alpha}_{i+1}\boldsymbol{C}_{\boldsymbol{x}_0})^{-1} \sqrt{\alpha_{i+1}} \\
&= \sqrt{\bar{\alpha}_T}((1 - \bar{\alpha}_0)\boldsymbol{I} + \bar{\alpha}_0 \boldsymbol{C}_{\boldsymbol{x}_0})((1 - \bar{\alpha}_T)\boldsymbol{I} + \bar{\alpha}_T \boldsymbol{C}_{\boldsymbol{x}_0})^{-1}.
\end{aligned} \tag{44}$$

For the second part, $\forall i = 1 : T - 1$, we have

$$\begin{aligned}
\boldsymbol{V}_{i-1,p_3}\boldsymbol{V}_{i,p_4} + \boldsymbol{V}_{i-1,p_4} &= ((1 - \bar{\alpha}_{i-1})\boldsymbol{I} + \bar{\alpha}_{i-1}\boldsymbol{C}_{\boldsymbol{x}_0})(1 - \bar{\alpha}_i + \bar{\alpha}_i\boldsymbol{C}_{\boldsymbol{x}_0})^{-1} \\
&\quad \cdot \sqrt{\alpha_i}(1 - \alpha_{i+1})((1 - \bar{\alpha}_{i+1})\boldsymbol{I} + \bar{\alpha}_{i+1}\boldsymbol{C}_{\boldsymbol{x}_0})^{-1} \sqrt{\bar{\alpha}_i} \\
&\quad + (1 - \alpha_i)((1 - \bar{\alpha}_i)\boldsymbol{I} + \bar{\alpha}_i\boldsymbol{C}_{\boldsymbol{x}_0})^{-1} \sqrt{\bar{\alpha}_{i-1}} \\
&= (1 - \alpha_i\alpha_{i+1})\sqrt{\bar{\alpha}_{i-1}}((1 - \bar{\alpha}_{i+1})\boldsymbol{I} + \bar{\alpha}_{i+1}\boldsymbol{C}_{\boldsymbol{x}_0})^{-1}.
\end{aligned} \tag{45}$$

Similarly, we obtain

$$\begin{aligned}
&\boldsymbol{V}_{0,p_3}(\boldsymbol{V}_{1,p_3}(\cdots \boldsymbol{V}_{T-2,p_3}(\boldsymbol{V}_{T-1,p_4}) + \boldsymbol{V}_{T-2,p_4}) + \boldsymbol{V}_{1,p_4}) + \boldsymbol{V}_{0,p_4} \\
&= (1 - \alpha_0\alpha_1\cdots\alpha_T)\sqrt{\bar{\alpha}_0}((1 - \bar{\alpha}_T)\boldsymbol{I} + \bar{\alpha}_T\boldsymbol{C}_{\boldsymbol{x}_0})^{-1} \\
&= (1 - \bar{\alpha}_T)((1 - \bar{\alpha}_T)\boldsymbol{I} + \bar{\alpha}_T\boldsymbol{C}_{\boldsymbol{x}_0})^{-1}.
\end{aligned} \tag{46}$$

Thus, we have

$$\begin{aligned}
\boldsymbol{\mu}_0(\boldsymbol{x}_1, \boldsymbol{y}) &= \sqrt{\bar{\alpha}_T}((1 - \bar{\alpha}_0)\boldsymbol{I} + \bar{\alpha}_0\boldsymbol{C}_{\boldsymbol{x}_0})((1 - \bar{\alpha}_T)\boldsymbol{I} + \bar{\alpha}_T\boldsymbol{C}_{\boldsymbol{x}_0})^{-1}\boldsymbol{\mu}_T \\
&\quad + (1 - \bar{\alpha}_T)((1 - \bar{\alpha}_T)\boldsymbol{I} + \bar{\alpha}_T\boldsymbol{C}_{\boldsymbol{x}_0})^{-1}\mathbb{E}_{p(\boldsymbol{x}_0|\boldsymbol{y})}[\boldsymbol{x}_0]
\end{aligned} \tag{47}$$

and $\boldsymbol{\mu}_0 \to \mathbb{E}_{p(\boldsymbol{x}_0|\boldsymbol{y})}[\boldsymbol{x}_0]$ as $\bar{\alpha}_T \to 0$.

## C  PROOF OF PROPOSITION 2

We have

$$
\begin{aligned}
p(\boldsymbol{x}_k|\boldsymbol{x}_{k+1:T}, \boldsymbol{y}) &= \frac{p(\boldsymbol{x}_{k:T}, \boldsymbol{y})}{p(\boldsymbol{x}_{k+1:T}, \boldsymbol{y})} \\
&= \frac{\int p(\boldsymbol{x}_0)p(\boldsymbol{y}|\boldsymbol{x}_0)p(\boldsymbol{x}_{k:T}|\boldsymbol{x}_0)d\boldsymbol{x}_0}{\int p(\boldsymbol{x}_0)p(\boldsymbol{y}|\boldsymbol{x}_0)p(\boldsymbol{x}_{k+1:T}|\boldsymbol{x}_0)d\boldsymbol{x}_0} \\
&= \frac{p(\boldsymbol{x}_{k+1:T}|\boldsymbol{x}_k)\int p(\boldsymbol{x}_0)p(\boldsymbol{y}|\boldsymbol{x}_0)p(\boldsymbol{x}_k|\boldsymbol{x}_0)d\boldsymbol{x}_0}{p(\boldsymbol{x}_{k+2:T}|\boldsymbol{x}_{k+1})\int p(\boldsymbol{x}_0)p(\boldsymbol{y}|\boldsymbol{x}_0)p(\boldsymbol{x}_{k+1}|\boldsymbol{x}_0)d\boldsymbol{x}_0} \\
&= \frac{p(\boldsymbol{x}_{k+1}|\boldsymbol{x}_k)\int p(\boldsymbol{x}_0)p(\boldsymbol{y}|\boldsymbol{x}_0)p(\boldsymbol{x}_k|\boldsymbol{x}_0)d\boldsymbol{x}_0}{\int p(\boldsymbol{x}_0)p(\boldsymbol{y}|\boldsymbol{x}_0)p(\boldsymbol{x}_{k+1}|\boldsymbol{x}_0)d\boldsymbol{x}_0} \\
&= \frac{p(\boldsymbol{x}_k, \boldsymbol{x}_{k+1}, \boldsymbol{y})}{p(\boldsymbol{x}_{k+1}, \boldsymbol{y})} = p_k(\boldsymbol{x}_k|\boldsymbol{x}_{k+1}, \boldsymbol{y}).
\end{aligned}
\tag{48}
$$

We minimize the KL divergence between variational joint posterior $q(\boldsymbol{x}_{0:T}|\boldsymbol{y}) = q(\boldsymbol{x}_T|\boldsymbol{y})\prod_{k=T-1}^{0} q_k(\boldsymbol{x}_k|\boldsymbol{x}_{k+1}, \boldsymbol{y})$ and joint posterior $p(\boldsymbol{x}_{0:T}|\boldsymbol{y}) = p(\boldsymbol{x}_T|\boldsymbol{y})\prod_{k=T-1}^{0} p(\boldsymbol{x}_k|\boldsymbol{x}_{k+1:T}, \boldsymbol{y})$ to obtain the optimal variational distribution $q$:

$$
\begin{aligned}
\mathcal{F} &= \int q(\boldsymbol{x}_{0:T}|\boldsymbol{y}) \log \frac{q(\boldsymbol{x}_{0:T}|\boldsymbol{y})}{p(\boldsymbol{x}_{0:T}|\boldsymbol{y})} d\boldsymbol{x}_{0:T} \\
&= \sum_{k=T-1}^{0} \int q(\boldsymbol{x}_{0:T}|\boldsymbol{y}) \log \frac{q_k(\boldsymbol{x}_k|\boldsymbol{x}_{k+1}, \boldsymbol{y})}{p(\boldsymbol{x}_k|\boldsymbol{x}_{k+1:T}, \boldsymbol{y})} d\boldsymbol{x}_{0:T} \\
&= \sum_{k=T-1}^{0} \int q(\boldsymbol{x}_{k:T}|\boldsymbol{y}) \log \frac{q_k(\boldsymbol{x}_k|\boldsymbol{x}_{k+1}, \boldsymbol{y})}{p(\boldsymbol{x}_k|\boldsymbol{x}_{k+1:T}, \boldsymbol{y})} d\boldsymbol{x}_{k:T} \\
&= \sum_{k=T-1}^{0} \int q(\boldsymbol{x}_{k:T}|\boldsymbol{y}) \log \frac{q_k(\boldsymbol{x}_k|\boldsymbol{x}_{k+1}, \boldsymbol{y})}{p_k(\boldsymbol{x}_k|\boldsymbol{x}_{k+1}, \boldsymbol{y})} d\boldsymbol{x}_{k:T} \\
&= \sum_{k=T-1}^{0} \int q(\boldsymbol{x}_{k:k+1}|\boldsymbol{y}) \log \frac{q_k(\boldsymbol{x}_k|\boldsymbol{x}_{k+1}, \boldsymbol{y})}{p_k(\boldsymbol{x}_k|\boldsymbol{x}_{k+1}, \boldsymbol{y})} d\boldsymbol{x}_k d\boldsymbol{x}_{k+1} \\
&= \sum_{k=T-1}^{0} \int q(\boldsymbol{x}_{k+1}|\boldsymbol{y}) \int q_k(\boldsymbol{x}_k|\boldsymbol{x}_{k+1}, \boldsymbol{y}) \log \frac{q_k(\boldsymbol{x}_k|\boldsymbol{x}_{k+1}, \boldsymbol{y})}{p_k(\boldsymbol{x}_k|\boldsymbol{x}_{k+1}, \boldsymbol{y})} d\boldsymbol{x}_k d\boldsymbol{x}_{k+1}
\end{aligned}
\tag{49}
$$

where the last line follows from

$$
\begin{aligned}
q(\boldsymbol{x}_{k:k+1}|\boldsymbol{y}) &= \int q(\boldsymbol{x}_T|\boldsymbol{y}) \prod_{i=k}^{T-1} q_i(\boldsymbol{x}_i|\boldsymbol{x}_{i+1}, \boldsymbol{y}) d\boldsymbol{x}_{k+2:T} \\
&= \int q_k(\boldsymbol{x}_k|\boldsymbol{x}_{k+1}, \boldsymbol{y}) q_{k+1}(\boldsymbol{x}_{k+1}|\boldsymbol{x}_{k+2}, \boldsymbol{y}) q(\boldsymbol{x}_{k+2:T}|\boldsymbol{y}) d\boldsymbol{x}_{k+2:T} \\
&= q_k(\boldsymbol{x}_k|\boldsymbol{x}_{k+1}, \boldsymbol{y}) \int q_{k+1}(\boldsymbol{x}_{k+1}|\boldsymbol{x}_{k+2}, \boldsymbol{y}) q(\boldsymbol{x}_{k+2:T}|\boldsymbol{y}) d\boldsymbol{x}_{k+2:T} \\
&= q_k(\boldsymbol{x}_k|\boldsymbol{x}_{k+1}, \boldsymbol{y}) q(\boldsymbol{x}_{k+1}|\boldsymbol{y}).
\end{aligned}
\tag{50}
$$

Thus, the minimization of $\mathcal{F}$ is equivalent to the minimization of

$$
\mathcal{F}_k = \int q_k(\boldsymbol{x}_k|\boldsymbol{x}_{k+1}, \boldsymbol{y}) \log \frac{q_k(\boldsymbol{x}_k|\boldsymbol{x}_{k+1}, \boldsymbol{y})}{p_k(\boldsymbol{x}_k|\boldsymbol{x}_{k+1}, \boldsymbol{y})} d\boldsymbol{x}_k, \forall k = 0, 1, \cdots T - 1.
\tag{51}
$$

## D  GAUSSIAN MIXTURE MODEL

The data prior of $x_0$ is given by

$$
\begin{aligned}
p(x_0) &= \frac{1}{2}\mathcal{N}(x_0; \mu_1, v_1^2) + \frac{1}{2}\mathcal{N}(x_0; \mu_2, v_2^2) \\
&= \frac{1}{2}(p_1(x_0) + p_2(x_0)).
\end{aligned}
\tag{52}
$$

Then, the data score is

$$
\nabla_{x_0} \log p(x_0) = \frac{1}{2p(x_0)}\left(-p_1(x_0)\frac{x_0 - \mu_1}{\sigma_1^2} - p_2(x_0)\frac{x_0 - \mu_2}{\sigma_2^2}\right).
\tag{53}
$$

The measurement $y = ax_0 + \sigma_0\varepsilon_1$ and for VP diffusion, $\boldsymbol{x}_k = \sqrt{\bar{\alpha}_k}x_0 + \sqrt{1 - \bar{\alpha}_k}\varepsilon_2$ where $\varepsilon_1$ and $\varepsilon_2$ are i.i.d. from $\mathcal{N}(0, 1)$. Then,

$$
x_0 = \frac{x_k - \sqrt{1 - \bar{\alpha}_k}\varepsilon_2}{\sqrt{\bar{\alpha}_k}}
\tag{54}
$$

and

$$
y = a\frac{x_k - \sqrt{1 - \bar{\alpha}_k}\varepsilon_2}{\sqrt{\bar{\alpha}_k}} + \sigma_0\varepsilon_1.
\tag{55}
$$

Thus, $p(y|x_k) \sim \mathcal{N}(\frac{ax_k}{\sqrt{\bar{\alpha}_k}}, \frac{a^2(1-\bar{\alpha}_k)}{\bar{\alpha}_k} + \sigma_0^2)$ and the likelihood score

$$
\nabla_{x_k} \log p(y|x_k) = \frac{a}{\sqrt{\bar{\alpha}_k}}\frac{y - \frac{a}{\sqrt{\bar{\alpha}_k}}x_k}{\frac{a^2(1-\bar{\alpha}_k)}{\bar{\alpha}_k} + \sigma_0^2}.
\tag{56}
$$

## E  RMP IMPLEMENTATION DETAILS

### E.1  HYPERPARAMETERS SETTING

In the experiments presented in Tables 1 and 2, the step sizes $s_1$ and $\zeta$ for different tasks on the FFHQ dataset for VP/VE-RMP are set according to Tables 3 and 4, respectively. Additionally, we conducted a parameter sensitivity analysis for the hyperparameters $s_1 \in (0, 1]$ and $\zeta \in (0, 1]$ of VP/VE-RMP on SR tasks to demonstrate the process of selecting these parameters. The performance of VP-RMP, measured by various metrics in relation to the hyperparameters, is shown in Fig. 5. From these colormesh plots, we observe that the performance of RMP varies smoothly with respect to the hyperparameters. Therefore, we can choose the hyperparameters based on the colormesh.

Table 3: Hyper-parameters of VP-RMP for different tasks

| parameter | SR ($4\times$) | box | random | Gauss | motion | nonlinear | phase |
|---|---|---|---|---|---|---|---|
| $s_1$ | 0.9 | 0.6 | 0.6 | 0.9 | 0.9 | 0.8 | 0.5 |
| $\zeta$ | 0.15 | 0.3 | 0.3 | 0.5 | 0.5 | 0.8 | 0.5 |

Table 4: Hyper-parameters of VE-RMP for different tasks

| parameter | SR ($4\times$) | box | random | Gauss | motion | nonlinear | phase |
|---|---|---|---|---|---|---|---|
| $s_1$ | 0.1 | 0.05 | 0.1 | 0.05 | 0.05 | 0.05 | 0.05 |
| $\zeta$ | 0.07 | 0.1 | 0.25 | 0.15 | 0.15 | 0.15 | 0.35 |

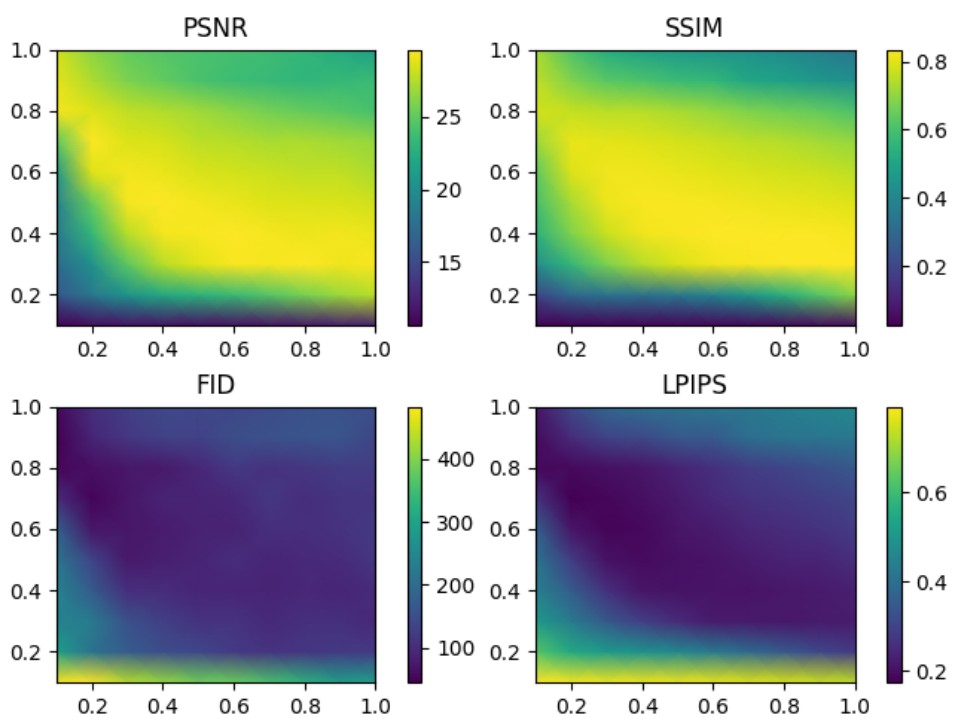

Figure 5: Hyperparameters ($s_1$ and $\zeta$) versus metrics for VP-RMP on FFHQ dataset.

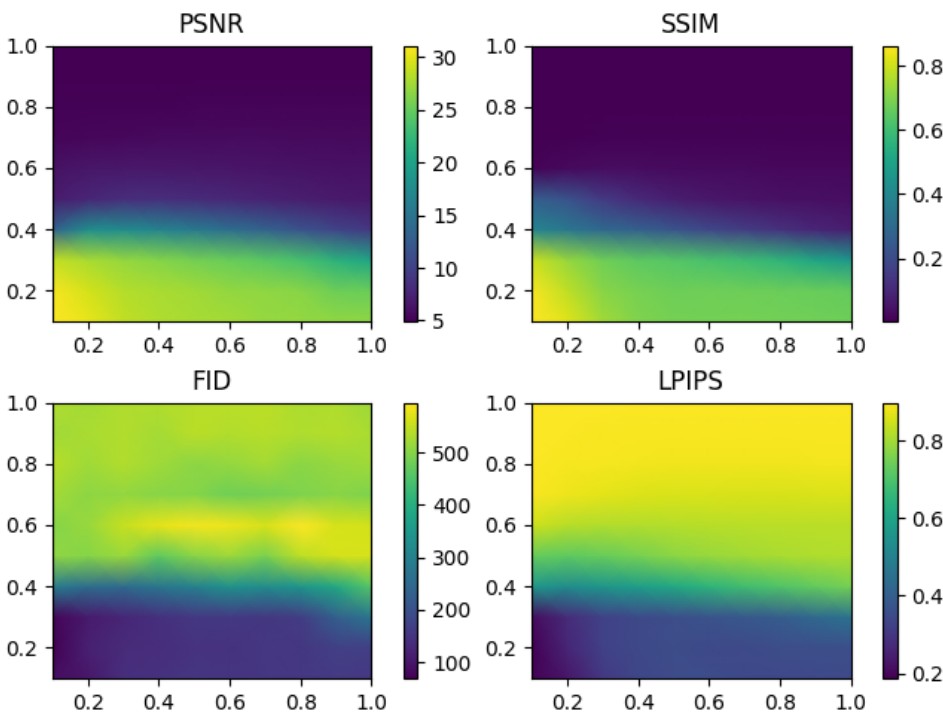

Figure 6: Hyperparameters ($s_1$ and $\zeta$) versus metrics for VE-RMP on FFHQ dataset.

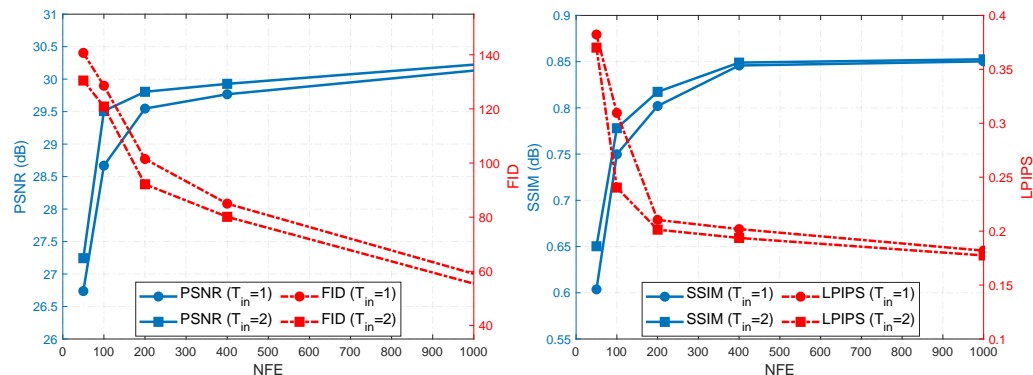

Figure 7: Performance comparison of VP-RMP with different $T$ and $T_{in}$ on SR task.

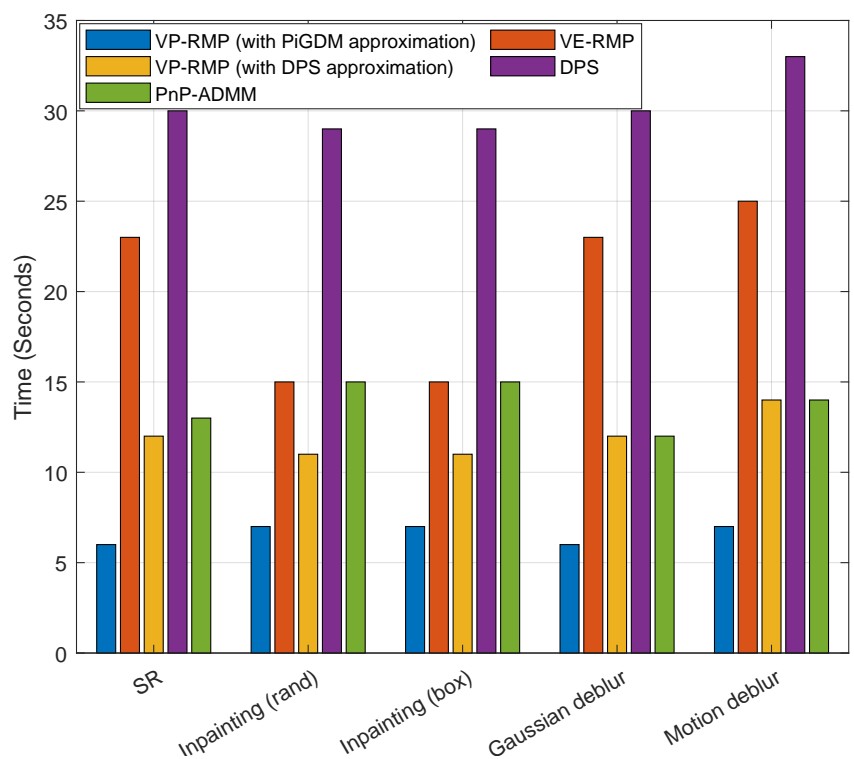

Figure 8: Running time of algorithms for different image reconstruction tasks.

### E.2 ALGORITHM PERFORMANCE AND RUNNING TIME COMPARISONS

To demonstrate the performance of RMP when choosing different values of $T$ and $T_{in}$, we present the performance of VP-RMP with respect to various numbers of loops in Fig. 7. From the figure, we observe that the performance of VP-RMP improves with an increasing number of outer loops $T$ and inner loops $T_{in}$. We compare the running times of VE-RMP, VP-RMP, DPS, and PnP-ADMM for different image reconstruction tasks on the FFHQ dataset in Fig. 8. From Fig. 8, it is evident that VP-RMP achieves better performance with shorter running times across all tasks.

### E.3 APPROXIMATION OF LIKELIHOOD SCORE FOR NATURAL IMAGES

In addition to the approximation of likelihood score we used in the main results for image reconstruction tasks, other approximation methods (Meng & Kabashima, 2022; Song et al., 2023) of likelihood score can be employed to RMP. We present the performance of VP-RMP with the approximation

method used in ΠGDM in Table 5. It is shown that the performance of VP-RMP with likelihood approximation method in Song et al. (2023) achieves similar performance as that of VP-RMP with DPS approximation.

Table 5: The performance of VP-RMPs with different approximations of likelihood score on FFHQ dataset.

| Methods | SR (4 ×) | | | |
|---|---|---|---|---|
| | PSNR | SSIM | FID | LPIPS |
| VP-RMP (with approximation in (19)) | 28.86 | 0.8413 | 59.7112 | 0.1825 |
| VP-RMP (with approximation in ΠGDM (Song et al., 2023)) | 28.70 | 0.8477 | 61.8101 | 0.1988 |

## F    IMAGE RECONSTRUCTION RESULTS

The image reconstruction results for inpainting (random) and Gaussian deblur tasks on FFHQ dataset are shown in Fig. 9 and more results of various image reconstruction tasks on ImageNet are presented in Fig. 10. The hyperparameters are set according to Table 6.

Table 6: Parameter settings of VP-RMP for different tasks on ImageNet.

| hyperparameter | SR (4×) | box | random | Gauss | motion | nonlinear | phase |
|---|---|---|---|---|---|---|---|
| $s_1$ | 0.6 | 0.6 | 0.6 | 0.6 | 0.6 | 0.8 | 0.5 |
| $\zeta$ | 0.2 | 0.9 | 0.9 | 0.9 | 0.9 | 0.8 | 0.5 |

## G    LIMITATIONS OF RMP

Although RMP achieves better performance with lower computational complexity compared to existing methods across a variety of inverse problems, as demonstrated by experiments, some limitations may restrict its application to general settings. As a plug and play method, RMP requires a pre-trained score network for the data. In our experiments, the pre-trained score networks were taken from previous works. Training the score network can be challenging for certain data distributions, and obtaining clean datasets can also be difficult in some fields. This is a common limitation shared by all score-based methods for inverse problems, such as DPS, DDRM, and ΠGDM. Additionally, the complexity of solving the variational problems using stochastic NGD, which is involved in updating the mean parameter in RMP, may hinder its application in real-time scenarios.

Table 7: Quantitative evaluation of algorithms on ImageNet dataset.

| Methods | SR (4 ×) | | Inpaint (box) | | Inpaint (random) | | Deblur (Gauss) | | Deblur (motion) | |
|---|---|---|---|---|---|---|---|---|---|---|
| | PSNR ↑ | SSIM ↑ | PSNR ↑ | SSIM ↑ | PSNR ↑ | SSIM ↑ | PSNR ↑ | SSIM ↑ | PSNR ↑ | SSIM ↑ |
| VP-RMP (Ours) | **26.21** | **0.7261** | **19.29** | **0.7830** | **29.71** | 0.8720 | **24.65** | 0.6806 | **25.30** | **0.6533** |
| DPS | 21.24 | 0.5655 | 17.96 | 0.7274 | 27.11 | 0.7862 | 22.49 | 0.5996 | 19.72 | 0.5118 |
| DDRM | 24.28 | 0.7137 | 19.01 | 0.7801 | 29.01 | **0.8742** | 23.81 | **0.7109** | - | - |
| Methods | SR (4 ×) | | Inpaint (box) | | Inpaint (random) | | Deblur(Gauss) | | Deblur (motion) | |
| | FID ↓ | LPIPS ↓ | FID ↓ | LPIPS ↓ | FID ↓ | LPIPS ↓ | FID ↓ | LPIPS ↓ | FID ↓ | LPIPS ↓ |
| VP-RMP (Ours) | **75.76** | **0.3021** | **106.73** | **0.2156** | **26.45** | 0.1400 | **85.92** | 0.3105 | **71.00** | **0.2933** |
| DPS | 187.14 | 0.4147 | 146.54 | 0.2664 | 76.20 | 0.2424 | 86.28 | 0.3224 | 144.43 | 0.3795 |
| DDRM | 117.82 | 0.3086 | 119.65 | 0.2232 | 27.18 | **0.1066** | 95.07 | **0.2931** | - | - |

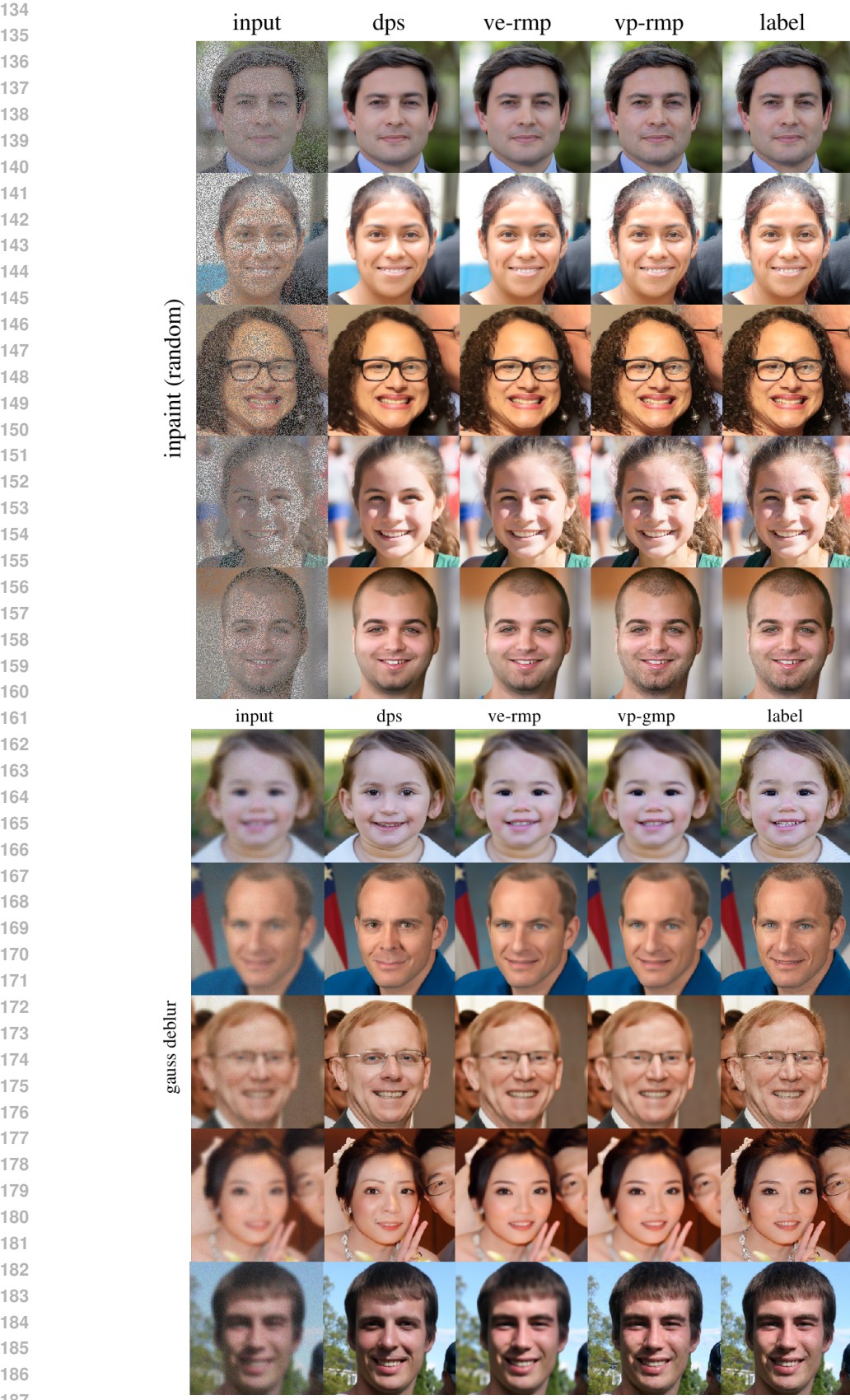

Figure 9: Inpainting (Random) and Gaussian deblur ($\epsilon = 0.05$).

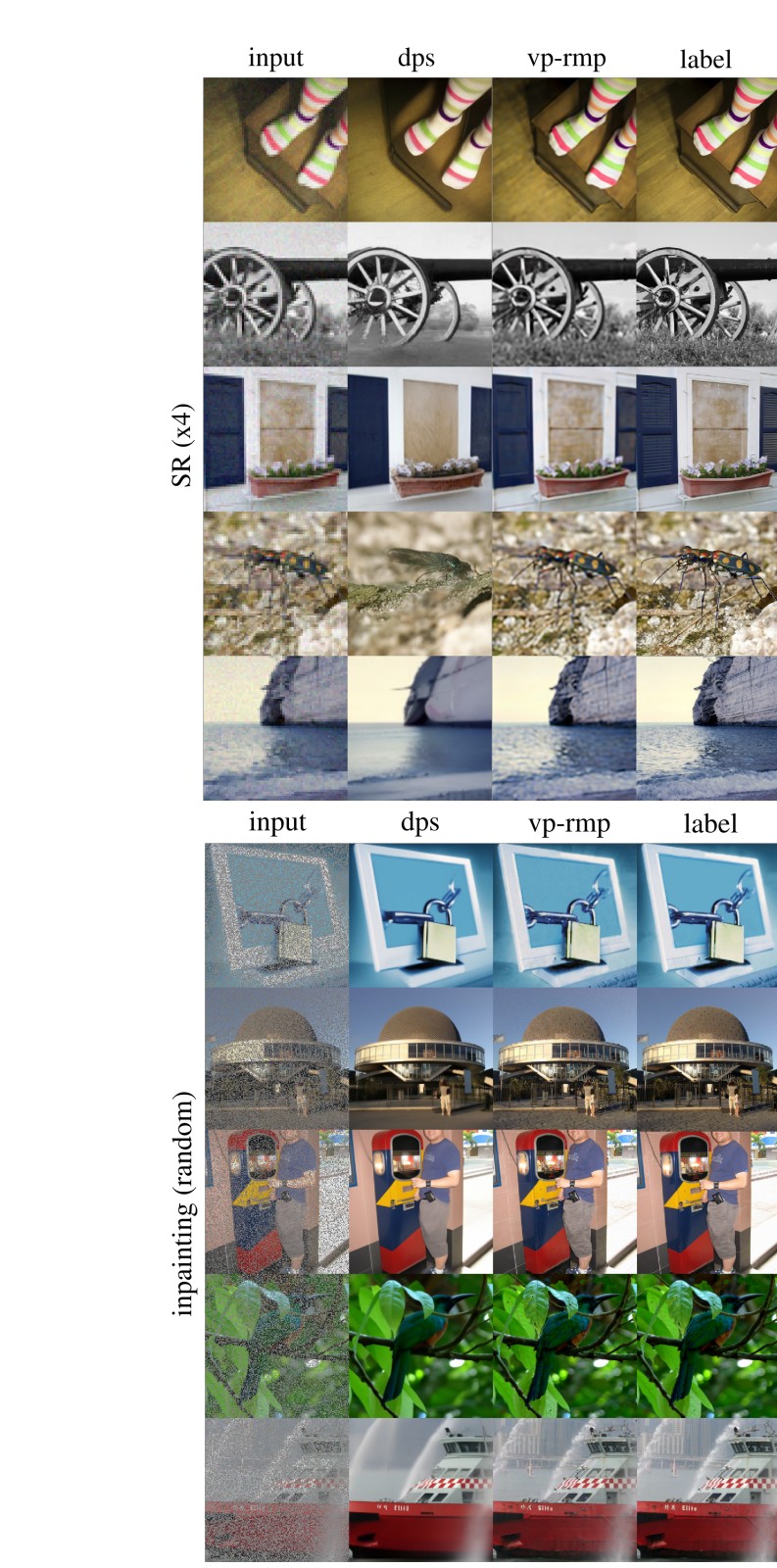

Figure 10: SR ($\times 4$) and Gaussian deblur ($\epsilon = 0.05$).

