# OpenReview forum: "Score-Based Variational Inference for Inverse Problems"
_ICLR.cc/2025/Conference — Submitted to ICLR 2025_

### Official Review · Reviewer_4zYw · 2024-10-25

**Soundness:** 3
**Presentation:** 3
**Contribution:** 3
**Rating:** 6
**Confidence:** 2

**Summary:**

This paper targets the estimating the posterior mean in the inverse problems.
To do this, the authors first argument the intractable posterior distribution with a diffusion process, and derive the mean and variance propagation rule in the backward diffusion process.
To estimate the intractable transitions incorporated in the above propagation rule, variational inference objectives as well as the natural gradient update are developed.
The above estimation relies on a pretrained score model on the clean data.
The experiments on toy examples and high-resolution images both demonstrate the effectiveness of the proposed method compared to baselines.

**Strengths:**

- This paper is well written and easy to follow.
- Although diffusion-based approaches for solving inverse problems are extensively investigated, the estimating strategy for the posterior mean in equation (7) is novel.
- The gradient estimation in equation (22) with a pre-trained score model is non-trivial. It proves that this method can provide reliable posterior mean estimation in the experiments.
- Figure 2 clearly illustrates how RMP propagates the mean through the diffusion process, which highlights the difference from the models targeting the distributional information.

**Weaknesses:**

- The citation formats, \citep and \citet, need to be more carefully distinguished.
- RMP requires an additional pre-trained score network to estimate the posterior mean. The evaluation in the experiments does not count the time consumption for training the score network. Moreover, the difficulty of obtaining such a score model can depend on the modeling objects, where images are only a special instance. If this is not possible, the authors should discuss this in their limitations.
- For each fixed measurement $y$, RMP requires two loops to obtain a posterior mean estimation. The outer loop iterates the timesteps in the diffusion process, and the  inner loop optimizes transition means by variational inference. Adding more computational cost comparison in experiments would help better evaluate the proposed method.
- For a different $y$, we need to re-run the expensive RMP to establish a posterior mean estimation. Especially, the $\mu$ and $\Lambda$ need to be re-estimated. Is an amortized inference approach possible here?

**Questions:**

- Is such an expensive two-loop circulation common in this field? If not, the authors should discuss this in their limitations. If so, a comparison with related works is recommended.

---

> ### Author Response · Authors · 2024-11-23
>
> **We thank reviewer 4zYw for careful reading and constructive comments.**
>
> W1: The citation formats, \citep and \citet, need to be more carefully distinguished.
>
>   > We appreciate the reviewer's comment and have addressed this issue in our manuscript.
>
> W2: RMP requires an additional pre-trained score network to estimate the posterior mean. The evaluation in the experiments does not count the time consumption for training the score network. Moreover, the difficulty of obtaining such a score model can depend on the modeling objects, where images are only a special instance. If this is not possible, the authors should discuss this in their limitations.
>
>   > RMP is a type of Plug and Play method that requires only a pre-trained score network for inference. The pre-trained score networks used for image reconstruction in our experiments are taken from previous works. It is fair not to include the training time of the score network in our comparisons, since all comparison methods require such a score network. To train such a score network, a large dataset is required, and denoising score matching is used for the training of the score network with the network architecture chosen, such as NCSN. For some applications, the clean dataset may be unavailable, which is a major obstacle. This limitation is common to all score-based methods used for inverse problems, including DPS, DDRM, and Pi-DGM. We have added a discussion on the limitations of RMP in the appendix of our manuscript.
>
> W3: For each fixed measurement $y$, RMP requires two loops to obtain a posterior mean estimation. The outer loop iterates the timesteps in the diffusion process, and the inner loop optimizes transition means by variational inference. Adding more computational cost comparison in experiments would help better evaluate the proposed method.
>
>   > We add VP-RMP with PiGDM approximation and PnP-ADMM to the running time comparison in Fig. 8 in Appendix E.2 of our manuscript. Also, we add a comparison of the performance of VP-RMP versus loop number in Fig. 7.
>
> W4: For a different $y$, we need to re-run the expensive RMP to establish a posterior mean estimation. Especially, the $\mu$ and $\Lambda$ need to be re-estimated. Is an amortized inference approach possible here?
>
>   > Our proposed RMP algorithm is a non-amortized variational inference method that necessitates the updating of $\mu$ and $\Lambda$ at each reverse step to accommodate distinct measurements $y$. Amortized variational inference learns a shared inference network that connects measurements to the posterior distribution of latent variables [1]. As shown in previous works [2], amortized variational inference incurs an amortization gap for general settings. Non-amortized methods typically perform better than amortized methods since the inference is based on the current measurement rather than relying on a generalized network for all measurements.
>   >
>   > To solve inverse problems using score-based data priors, we believe that amortized methods can be applied to applications with loose requirements for the solution, where the focus is more on computational complexity. This is because amortized methods reduce inference complexity and provide direct output estimation. This is an interesting topic that deserves more future research.
>   >
>   > [1] Ganguly, Ankush, Sanjana Jain, and Ukrit Watchareeruetai. "Amortized Variational Inference: A Systematic Review." *Journal of Artificial Intelligence Research* 78 (2023): 167-215.
>   >
>   > [2] Margossian, Charles C., and David M. Blei. "Amortized Variational Inference: When and Why?." *arXiv preprint arXiv:2307.11018* (2023).
>
> Q1: Is such an expensive two-loop circulation common in this field? If not, the authors should discuss this in their limitations. If so, a comparison with related works is recommended.
>
> > To the best of our knowledge, our proposed method, RMP, is the first to employ variational inference with a score-based prior for tracking the density evolution of the conditional reverse process, while the mean simultaneously converging on the posterior mean. The two-loop form is not commonly used in score-based methods for inverse problems. We have addressed this limitation in Appendix G of our manuscript.

---

> > ### Comment · Reviewer_4zYw · 2024-11-27
> > **Thanks!**
> >
> > The authors' response has adequately addressed my concerns. I will maintain the current score to reflect my positive view.

---

### Official Review · Reviewer_nsm7 · 2024-11-01

**Soundness:** 3
**Presentation:** 3
**Contribution:** 3
**Rating:** 6
**Confidence:** 4

**Summary:**

This paper presents a score-based variational inference method for inverse problems. The key ingredient is a framework called reverse mean propogation (RMP) that targets the posterior mean directly. The authors show that RMP can be implemented by solving a variational problem which decomposes into independent sub variational problems at each reverse step. Natural gradient descent is used for learning variational parameters. The authors show that the proposed algorithm outperforms state-of-the-art algorithms in various inverse problems.

**Strengths:**

1. The paper is clearly written and estimating the posterior mean directly seems new for score-based inverse problems.
2. The authors show that in the limit of continous time, the end point of reverse mean propogation would recover the exact posterior mean, with different conditions for VP and VE diffusions respectively.
3. Natural gradient descent is employed for optimization.
4. The authors have evaluated their algorithms thoroughly, and also conducted an ablation study on the complexity vs performance.

**Weaknesses:**

1. Although the derivation of the proposed algorithm seems right, there are lots of approximations that would damage the validity of the methods. For example, proposition 1 and theorem 1 only hold in the continuous limit, not sure how accurate it would be, especially when the step size is large (or equivalently, $T$ is small). Can the authors provide an empirical analysis or theoretical bounds on the approximation error for finite T? For example, can the authors show how the algorithm's performance changes as T varies, to demonstrate how closely it approaches the continuous limit in practice?

2. It seems that the proposed method also uses the same approximation as in DPS for the likelihood score (section 4.3). As this is the main reason that DPS would perform poorly, can the author explain why your algorithm would perform better than DPS? any intuition? Can the authors provide a detailed comparison or ablation study isolating the impact of the likelihood score approximation versus other aspects of their approach?

3. There are some hyperparameters (i.e., $\gamma_k$ in equation (22) and the stepsize $s_1$) that would be crucial for the performance. The paper lacks ablation studies on it. Can the authors conduct and report sensitivity analyses for $\gamma_k$ and $s_1$, showing how performance metrics change across reasonable ranges for these parameters?

**Questions:**

1. It seems that the overall complexity is the product of the number of the outer loops $T$ and the number of the inner loops $T_{in}$. Would different choices of these numbers affect the performance? For example, one can make $T$ larger, in the hope that the correspondng posterior $p(x_k|x_{k+1},y)$ would become easier to approximate so that the convergence would be faster for VI and $T_{in}$ can therefore be smaller.

2. It seems that VE-RMP tends to require smaller $T$, can the author explain it?

---

> ### Author Response · Authors · 2024-11-23
>
> **We thank reviewer nsm7 for careful reading and constructive comments.**
>
> W1. Although the derivation of the proposed algorithm seems right, there are lots of approximations that would damage the validity of the methods. For example, proposition 1 and theorem 1 only hold in the continuous limit, not sure how accurate it would be, especially when the step size is large (or equivalently, $T$ is small). Can the authors provide an empirical analysis or theoretical bounds on the approximation error for finite $T$? For example, can the authors show how the algorithm's performance changes as $T$ varies, to demonstrate how closely it approaches the continuous limit in practice?
>
>    > Our proposed RMP algorithm is essentially a score-based implementation of the Forker-Planck equation [1] for a conditional reverse process that tracks the evolution of the probability density of that process. RMP approximates this evolution using the reverse mean and covariance, as presented in Proposition 1 of our manuscript. The implementation of RMP, based on variational inference, serves as an approximation method for the discretization  of the evolution process. This approximation error for finite $T$ is generally an open question since $p(x\_0|y)$ is generally intractable. The RMP algorithm presented in Algorithm 2 is a discrete realization of the RMP framework, meaning that RMP approaches the posterior mean strictly as T tends to infinity. The theoretical performance of RMP when the $T$ is finite can be characterized by the variational loss. We next give an intuition on the theoretical bound of approximation error. From the discrete variational loss F defined in (49), a continuous loss can also be defined similarly. The theoretical performance of RMP when T is finite can be characterized by the variational loss. Next, we will provide an intuition regarding the theoretical bound on the approximation error. Similarly, from the discrete variational loss F defined in (49), a continuous loss can also be defined. The difference between the discrete loss and the continuous loss can be interpreted as the approximation error bound. The continuous loss can be upper bounded by the discrete loss using Jensen's inequality, and Jensen's gap also provides the approximation bound. We have added an experiment in the appendix of our manuscript to demonstrate the performance trend of VP-RMP compared to the outer loop number $T$, as shown in Fig. 7.
>    >
>    > [1] Risken, H. "The Fokker-Planck Equation." (1996).
>
> W2. It seems that the proposed method also uses the same approximation as in DPS for the likelihood score (section 4.3). As this is the main reason that DPS would perform poorly, can the author explain why your algorithm would perform better than DPS? any intuition? Can the authors provide a detailed comparison or ablation study isolating the impact of the likelihood score approximation versus other aspects of their approach?
>
>    > Different from the previous works that draw samples from the posterior distribution of data using a SDE, our proposed RMP algorithm is essentially a score-based implementation of the Fokker-Planck equation for a conditional reverse process that tracks the evolution of the probability density of that process. RMP approximates this evolution using the reverse mean and covariance. The implementation of RMP, based on variational inference, serves as an approximation method for the discretization of the evolution process. This represents a fundamental difference between our method and previous sampling-based methods. We demonstrate this difference with the following example: We set the parameters of VP-RMP as follows: $T\_{in} = 1$, step size $s1 = 1/\alpha\_{k+1}$, and $x\_k = \mu\_k$ (which is not sample from the variational distribution). In this case, RMP reduces to DPS. Therefore, the reason RMP performs better than DPS is that at each reverse step, DPS samples from the posterior, while RMP tracks the evolution of the density using the mean and covariance (approximated by precision), which leads to better performance in PSNR.
>
> W3. There are some hyperparameters (i.e., 𝛾𝑘 in equation (22) and the stepsize 𝑠1) that would be crucial for the performance. The paper lacks ablation studies on it. Can the authors conduct and report sensitivity analyses for 𝛾𝑘 and 𝑠1, showing how performance metrics change across reasonable ranges for these parameters?
>
>    > We conduct a parameter sensitivity analysis for the hyperparameters stepsize $s_1$ and $\gamma_k$ involved in the RMP algorithm. In Figs. 5 and 6 of Appendix E.1, the performance of VP-RMP and VE-RMP versus hyperparameters, in terms of four metrics, is shown in the form of a 2D colormesh. From the colormesh, we see that all four metrics exhibit similar variation when the hyperparameters change, and these metrics vary smoothly with respect to the hyperparameters. Therefore, we can choose the hyperparameters based on the colormesh.

---

> > ### Author Response · Authors · 2024-11-23
> >
> > Q1. It seems that the overall complexity is the product of the number of the outer loops $𝑇$ and the number of the inner loops $𝑇_{𝑖𝑛}$. Would different choices of these numbers affect the performance? For example, one can make $𝑇$ larger, in the hope that the corresponding posterior $𝑝(𝑥\_𝑘|𝑥\_{𝑘+1},𝑦)$ would become easier to approximate so that the convergence would be faster for VI and $𝑇\_{in}$ can therefore be smaller.
> >
> >    > It is true that the choices of $𝑇$ and $𝑇\_{in}$ affect the performance of RMP. As shown in Fig. 7 in the appendix of our manuscript, we observe that as the outer loop $𝑇$ increases, the performance of RMP improves, while the $𝑇\_{in}$ required to achieve a similar level of performance decreases.
> >
> > Q2. It seems that VE-RMP tends to require smaller $𝑇$ , can the author explain it?
> >
> >    > In VE-RMP, the noise added to the original data in the forward process scales quickly, and the step size of NGD used in VE-RMP is set according to the noise variance, leading to faster convergence in $𝑇$. It is also important to note that although $𝑇$ in VE-RMP is smaller, it requires a larger $𝑇\_{in}$ to achieve performance comparable to that of VP-RMP.

---

> ### Comment · Reviewer_nsm7 · 2024-11-29
>
> Thanks for the response and additional results. I will maintain my score.
>
> Regarding your discussion on the connection to DPS, the coefficient before term $\nabla \log p(y|x_k)$ is set to 1 in DPS, while in your case it is set as a hyperparameter $\gamma_k$. What would DPS perform if one also treat this coefficient as a hyperparameter (just as you did in RMP), instead of fixing at 1?

---

> > ### Author Response · Authors · 2024-11-29
> >
> > Dear reviewer nsm7, just as there is a hyperparameter $\gamma_k$ in RMP, there is also a hyperparameter $\zeta_k$ in DPS that controls the likelihood score approximation (see page 6 of theri papar). The parameter setting for $\zeta_k$ is provided in their appendix and also in their code.

---

> > > ### Comment · Reviewer_nsm7 · 2024-11-29
> > >
> > > So you just follow DPS for this balancing operation? Please cite their paper in the right place as well. Thanks for the clarification. I really like the idea of using guided diffusion for inverse problem. But all these tricks involved makes it hard to understand how it really works.

---

> > > > ### Author Response · Authors · 2024-11-29
> > > >
> > > > Dear reviewer nsm7,
> > > >
> > > > Thank you for your careful reading. We further clarify the setting of hyperparameter as follows.
> > > >
> > > > The idea of balancing the likelihood score is common. However, the strategy for setting the parameter is different.
> > > >
> > > > In RMP, we set the parameter as $\gamma_k = \zeta \frac{\|s_\theta(x_k,\sigma_k)\|_2}{\|\nabla \log p(y|\hat{x_0}(x_k))\|_2}$, where $\zeta$ is a constant.
> > > >
> > > > In DPS, the parameter is set as $\zeta_k =  \frac{ \zeta'}{\|\nabla \log p(y|\hat{x_0}(x_k))\|_2}$ where $\zeta'$ is a constant. For example, as provide in their paper, for Gaussian deblur task on ImageNet, the $\zeta'$ of DPS is set to 0.4.
> > > >
> > > > In our experiments, the implementation of DPS follows the code provided by the authors. We will include the above discussion in our paper after the acceptance.

---

### Official Review · Reviewer_ZpSm · 2024-11-02

**Soundness:** 3
**Presentation:** 3
**Contribution:** 3
**Rating:** 6
**Confidence:** 4

**Summary:**

This paper introduces a novel framework called Reverse Mean Propagation (RMP) for estimating the posterior mean. The framework involves tracking the mean and covariance in the evolution of the conditional reverse diffusion process.
This is achieved by minimizing the reverse KL divergence at each step through a variational inference with stochastic natural gradient descent.
In experiments, the authors validate their method on image reconstruction tasks. The results show that RMP outperforms than the existing baselines, including DPS and MCG methods.

**Strengths:**

S1: The paper proposes a novel framework, Reverse Mean Propagation, which significantly reduces the complexity of estimating the posterior mean compared to other methods that rely on generating samples from the posterior distribution.

S2: The details of each component of the RMP algorithm are well presented, including the reverse mean updates and estimation using stochastic natural gradient descent. The discussion on estimating the trace of the Hessian matrix $\nabla^2_{x_k} \log p_k$ is also clear and informative.

**Weaknesses:**

W1: In line 148, the authors discuss the variance preserving (VP) diffusion model and the variance exploding (VE) diffusion model, mentioning their different training approaches. It would be helpful to explain that the VP scheme, like the VE scheme, is also equivalent to learning the score functions of perturbed data distributions[1]. Clarifying this point could provide more comprehensive background information.

W2: In Table 2, the FID results appear inconsistent with the previously reported results of the DPS method in Table 4 of [2].

W3: The comparison methods in the paper are relatively limited. Including additional baselines, such as PnP-ADMM [3], would be beneficial.

W4: As a specific type of VI method, there is limited discussion on previous works that directly estimate the posterior mean.

[1] Song, Yang, et al. "Score-based generative modeling through stochastic differential equations." ICLR 2021.

[2] Chung, Hyungjin, et al. "Diffusion posterior sampling for general noisy inverse problems." ICLR 2023.

[3] Chan, Stanley H., Xiran Wang, and Omar A. Elgendy. "Plug-and-play ADMM for image restoration: Fixed-point convergence and applications." IEEE Transactions on Computational Imaging 3.1 (2016): 84-98.

**Questions:**

Q1: In line 703, the authors present the estimated form $p(x_k|y) \approx \mathcal{N}(x_k; \mathbb{E}\_{p(x_0|y)}[x_0], (\sigma_k^2 I + C_{x_0}))$.
However, $x_k|y$ is not generally an asymptotically Gaussian distribution when $\Delta t \rightarrow 0$. Could the authors provide further clarification on this estimation?

---

> ### Author Response · Authors · 2024-11-23
>
> **We thank reviewer ZpSm for careful reading of our work and the positive comments.**
>
> **Reply to reviewer's comments:**
>
> W1: In line 148, the authors discuss the variance preserving (VP) diffusion model and the variance exploding (VE) diffusion model, mentioning their different training approaches. It would be helpful to explain that the VP scheme, like the VE scheme, is also equivalent to learning the score functions of perturbed data distributions[1]. Clarifying this point could provide more comprehensive background information.
>
> > We thank Reviewer ZpSm for the valuable advice. We have clarified this point in Section 2.3.
>
> W2: In Table 2, the FID results appear inconsistent with the previously reported results of the DPS method in Table 4 of [2].
>
> > The inconsistency arises from the fact that most of the recovery results of DPS differ significantly from the original image, which indicates a high rate of unsuccessful recovery, as discussed in the Limitations section of the DPS paper. The metrics reported by DPS for phase retrieval in their paper only account for successful recovery. Additionally, we note that the comparison results of DPS were obtained by running the code provided by the authors, and the parameters of DPS were set according to their code instructions.
>
> W3: The comparison methods in the paper are relatively limited. Including additional baselines, such as PnP-ADMM [3], would be beneficial.
>
> > We have added PnP-ADMM for comparison in our manuscript. The implementation of PnP-ADMM is from the scico library. The parameters for PnP-ADMM are set to $\rho=0.2$ and $\text{maxiter}=12$, and the image denoiser is chosen as the pretrained DnCNN denoiser. From the comparison results, we observe that RMP significantly outperforms PnP-ADMM in all metrics.
>
> W4: As a specific type of VI method, there is limited discussion on previous works that directly estimate the posterior mean.
>
> > In this work, we present an evolution analysis of the transition probability density of the conditional reverse diffusion process, which is described by the Fokker-Planck Equation [1,2]. Different from the previous works that draw samples from the posterior distribution of data using a stochastic differential equation (SDE), our proposed RMP algorithm is essentially a score-based implementation of the Fokker-Planck equation for a conditional reverse process that tracks the evolution of the probability density of that process. RMP approximates this evolution using the reverse mean and covariance. The implementation, based on variational inference, serves as an approximation method for the discretization of the evolution process. This represents a fundamental difference between our method and previous sampling-based methods.
> >
> > We have added a discussion on related works in the introduction of our manuscript. Please see the highlighted part in the introduction.
> >
> > [1] Risken, H. "The Fokker-Planck Equation." (1996)
> >
> > [2] Jordan, Richard, David Kinderlehrer, and Felix Otto. "The variational formulation of the Fokker-Planck equation." *SIAM journal on mathematical analysis* 29.1 (1998): 1-17.
>
> Q1: In line 703, the authors present the estimated form $𝑝(𝑥\_𝑘|𝑦) \approx \mathcal{N}(𝑥\_𝑘;𝐸 𝑝(𝑥\_0|𝑦)[𝑥\_0],(\sigma\_𝑘^2 𝐼+𝐶\_{𝑥\_0}))$. However, $p(𝑥\_𝑘|𝑦)$ is not generally an asymptotically Gaussian distribution when $\Delta 𝑡\rightarrow 0$. Could the authors provide further clarification on this estimation?
>
> > It is true that $p(x_k|y)$ is generally not a Gaussian distribution as $\Delta t \to 0$. In Appendix A.2, we aim to calculate the mean and covariance of $p(x_k|x_{k+1},y)$, which is asymptotically Gaussian as $\Delta t \to 0$ [1,2]. Since $p(x_k|x_{k+1})$ is also Gaussian, we can approximate $p(x_k|y)$ as Gaussian when $\Delta t \to 0$ in the calculation.
> >
> > [1]Anderson, Brian DO. "Reverse-time diffusion equation models." *Stochastic Processes and their Applications* 12.3 (1982): 313-326.
> >
> > [2] Sohl-Dickstein, Jascha, et al. "Deep unsupervised learning using nonequilibrium thermodynamics." *International conference on machine learning*. PMLR, 2015.

---

> > ### Comment · Reviewer_ZpSm · 2024-11-26
> >
> > I would like to thank the authors for their detailed response. The experimental results now seem more convincing and I have raised my confidence score. It would be better if a discussion of other direct methods for estimating the posterior mean were included. However, this is just a minor point.

---

### Official Review · Reviewer_k8qn · 2024-11-03

**Soundness:** 3
**Presentation:** 4
**Contribution:** 3
**Rating:** 6
**Confidence:** 4

**Summary:**

This paper tackles the challenge of solving inverse problems using diffusion models and introduces a framework called Reverse Mean Propagation (RMP) to recover the posterior mean of the latent variable ${\bf x} _0$ given the measurement ${\bf y}$. RMP is based on the insight that the reverse conditional distributions $p({\bf x} _k|{\bf x} _{k+1},{\bf y})$ in diffusion models are approximately Gaussian, with their mean and variance depending on the unknown quantities $\mathbb{E}({\bf x} _0|{\bf y})$ and ${\rm Cov}({\bf x} _0|{\bf y})$. The paper proposes a Gaussian variational inference approach to learn the reverse conditional mean and approximate the reverse conditional variance.

**Strengths:**

The paper focuses on predicting $\mathbb{E}({\bf x} _0|{\bf y})$ rather than sampling from the distribution $p({\bf x} _0|{\bf y})$, which sets it apart from most existing works. The Gaussian VI approach for learning the reverse conditional mean is innovative, and the experimental results presented are also compelling. The writing is clear, and the mathematical derivations are solid. Overall, this paper is of high quality and meets the standards of the ICLR conference.

**Weaknesses:**

The introduction section would benefit from a concise summary of the main contributions. Additionally, a detailed literature review of existing methods for inverse problems, particularly those related to VI, should be included to highlight the novelty of this approach.

In the experiments, the proposed method RMP is compared with DPS, MCG, DDRM, and $\Pi$GDM. While RMP demonstrates strong performance against these baselines, it is important to note that most of them (DPS, MCG, and $\Pi$GDM) do not require additional training, and can be used in a plug-and-play manner with any pretrained diffusion model and suitable measurement process ${\bf y}={\cal A}({\bf x} _0)+{\bf w} _0$. However, RMP incurs extra overhead due to the stochastic NGD update for Gaussian VI. Furthermore, in section 5.3, the comparison of NFE between RMP and DPS focuses on inference time, and does not account for training complexity, which may render the comparison a little bit unfair.

**Questions:**

1. I'm trying to understand proposition 1 from a continuous perspective, as it derives the continuous-time limit of the distribution $p({\bf x} _k|{\bf x} _{k+1},{\bf y})$. However, it appears that these two approaches differ significantly.

   Consider the forward process $d{\bf x} _t=\sqrt{\frac{d}{dt}(\sigma _t^2)}d{\bf B} _t$, $t\in[0,1]$, ${\bf x} _0\sim p({\bf x} _0|{\bf y})$. We know that $p({\bf x} _t|{\bf x} _0)={\cal N}({\bf x} _0,\sigma _t^2{\bf I})$, and the marginal distribution of ${\bf x} _t$ is $p _t({\bf x} _t|{\bf y})$. The backward process is $d{\bf x} _t=-\frac{d}{dt}(\sigma _t^2)\nabla\log p _t({\bf x} _t|{\bf y})dt+\sqrt{\frac{d}{dt}(\sigma _t^2)}d{\bf B}^\gets _t$, where ${\bf B}^\gets _t$ is a backward BM. If initialized at ${\bf x} _1\sim p _1({\bf x} _1|{\bf y})$, then ${\bf x} _0\sim p _0({\bf x} _0|{\bf y})$. We can express the score as $\nabla\log p _t({\bf x} _t|{\bf y})=\frac{1}{\sigma _t^2}(\mathbb{E}({\bf x} _0|{\bf x} _t,{\bf y})-{\bf x} _t)$, and thus the discretization scheme can be written as ${\bf x} _{t-\Delta t}\approx {\bf x} _t+\frac{\sigma _t^2-\sigma _{t-\Delta t}^2}{\sigma _t^2}(\mathbb{E}({\bf x} _0|{\bf x} _t,{\bf y})-{\bf x} _t)+{\cal N}(0,(\sigma^2 _t-\sigma^2 _{t-\Delta t}){\bf I})$. (I hope my calculation is correct.)


   Compare this with proposition 1, the posterior variance is independent of ${\bf y}$, while the posterior mean involves $\mathbb{E}({\bf x} _0|{\bf x} _t,{\bf y})$ rather than $\mathbb{E}({\bf x} _0|{\bf y})$. Could you clarify the differences and connections between these two approaches?

2. I noticed an extra term $N\Lambda _k$ in line 2 of equations 17 and 18, which does not appear in equation 16. Is this an error? Additionally, since we are performing gradient descent, the plus and minus signs in equation 17 seem incorrect after the "=" sign.

3. Minor comments:

- Please use parenthetical citation `\citep{}` instead of `\cite{}` when the authors' names aren't part of the text. For example, the first sentence should be "Diffusion models (Sohl-Dickstein et al. 2015; Song & Ermon 2019; Ho et al. 2020; Song et al. 2020a; Rombach et al. 2022) have shown impressive performance for image generation."

- In proposition 1, the reverse conditional should be *approximately* Gaussian when $\Delta t\to0$. $k=0\cdots T-1$ should be $k=0,\cdots,T-1$.

- In line 234: $p({\bf x} _k|{\bf x} _{k+1:T},{\bf y})$ should be $p({\bf x} _k|{\bf x} _{k+1},{\bf y})$.

- Throughout the paper, $KL$ should be ${\rm KL}$, and $Cov$ should be ${\rm Cov}$.

- In equation 15, $\nabla _{{\bm\Lambda} _k}$ should be $\nabla _{{\Lambda} _k}$.

- In line 287, there is no need to mention $\Lambda _k=v _k^{-1}$.

- In algorithm 2, the roles of $T _s$ and $T _{in}$ need to be stated.

- I suggest the author(s) replace figures 1 and 2 with higher resolution versions, especially the bottom-right in figure 2.

- The title of section 6 has a spelling error.

---

> ### Author Response · Authors · 2024-11-23
>
> **We thank reviewer k8qn for carefully reading and constructive comments**
>
> W1: The introduction section would benefit from a concise summary of the main contributions. Additionally, a detailed literature review of existing methods for inverse problems, particularly those related to VI, should be included to highlight the novelty of this approach.
>
> > We thank reviewer k8qn for the constructive comments. We have added a discussion of our contributions and a literature review of existing methods for inverse problems and their relation to VI in the introduction. Please see our manuscript.
>
> W2: In the experiments, the proposed method RMP is compared with DPS, MCG, DDRM, and ΠGDM. While RMP demonstrates strong performance against these baselines, it is important to note that most of them (DPS, MCG, and ΠGDM) do not require additional training, and can be used in a plug-and-play manner with any pretrained diffusion model and suitable measurement process 𝑦=𝐴(𝑥0)+𝑤0. However, RMP incurs extra overhead due to the stochastic NGD update for Gaussian VI. Furthermore, in section 5.3, the comparison of NFE between RMP and DPS focuses on inference time, and does not account for training complexity, which may render the comparison a little bit unfair.
>
> > Similar to DPS, MCG, and ΠGDM, the proposed method RMP is also a plug and play method that does not require additional training if a pretrained score network is available. With a pretrained score network, RMP works well for suitable measurement process since the stochastic NGD involved in RMP updates the mean based on the natural gradient approximated by score networks. Therefore, the comparison of inference times between our method and DPS is fair, as no extra training is required for our method.
>
> Q1:
> > We thank reviewer k8qn for careful reading and constructive comment. Your calculation is completely correct and can lead to an alternative way to understand Proposition 1 if we take one step further. Specifically, if we replace the score ∇log⁡𝑝𝑡(𝑥𝑡|𝑦) using the Gaussian approximation with the mean and variance given by (31) and (32), i.e., $ \nabla log⁡ 𝑝\_𝑡(𝑥\_𝑡|𝑦) =(\sigma\_k^2 I + C\_{x\_0})^{-1}(E[x\_0|y] - x\_t) $ and take expactation, we arrive at the VE case of Proposition 1. It worth noting that the Gaussian approximation holds when $\Delta t\rightarrow 0$, which comes from the Gaussian property of the reverse diffusion process [1,2].
>
> [1]Anderson, Brian DO. "Reverse-time diffusion equation models." *Stochastic Processes and their Applications* 12.3 (1982): 313-326.
>
> [2] Sohl-Dickstein, Jascha, et al. "Deep unsupervised learning using nonequilibrium thermodynamics." *International conference on machine learning*. PMLR, 2015.
>
> Q2: I noticed an extra term $𝑁\Lambda\_𝑘$ in line 2 of equations 17 and 18, which does not appear in equation 16. Is this an error? Additionally, since we are performing gradient descent, the plus and minus signs in equation 17 seem incorrect after the "=" sign.
>
> > Dear Reviewer k8qn, this is not an error. Equation (16) represents the natural gradient expression of the loss function defined in Line 278, and Equations (17) and (18) represent the natural gradient updates of the parameters based on the loss function defined in (14). Additionally, the $𝑁\Lambda_𝑘$ term comes from the natural gradient of the first part of loss function in equation (14), specifically the gradient of $-N/2\log(2\pi/\Lambda_𝑘)$. Furthermore, the minus signs in equation 17 arise from the loss function in equation (14).
>
> Q3: We thank the reviewer for carefully reading. We have correct these problems in our manuscript.

---

> > ### Comment · Reviewer_k8qn · 2024-11-25
> >
> > I would like to thank the authors for making a detailed point-to-point response to my review, and am pleased to see the improvement in the revised version of the manuscript. I'm sorry about my misunderstanding on the plug-and-play nature of this sampler, and I suggest that the authors emphasize this in the main text to highlight this contribution.
> >
> > I'm still a little bit confused about your response to Q1. My main question is how are these two derivations related. I understand that your $\nabla\log p _ t ({\bf x} _ t |y)$ is derived from approximating it as a Gaussian, so that the update ${\bf x} _ {(k+1)\Delta t}\to {\bf x} _ {k\Delta t}$ is also a Gaussian distribution. On the other hand, in my derivation from the continuous perspective, when we also discretize the SDE, then the update ${\bf x} _ {(k+1)\Delta t}\to {\bf x} _ {k\Delta t}$ can be viewed as a different Gaussian distribution. These two distributions have different means and covariance, and I'm just wondering that when $\Delta t\to0$, how are these two Gaussian distributions related? Would they become equivalent in some sense? This would be a critical question to theoretically characterize the discretization error in the RMP algorithm.
> >
> > Except from this issue, I currently do not have further question on the paper.

---

> > > ### Author Response · Authors · 2024-11-27
> > >
> > > We thank the reviewer for the constructive insight. The derivation of reviewer k8qn is equivalent to our result when $\Delta t \rightarrow 0$. We next provide the justification.
> > >
> > > The reverse SDE of a reverse VE diffusion process is given by $d x\_t =-\frac{d }{dt}(\sigma\_t^2) \nabla_{t}\log p\_t(x\_t|y)+\sqrt{\frac{d}{dt}(\sigma\_t^2)}dB\_t$. Then, when $\Delta t\rightarrow 0$, we have, $x\_{k \Delta t} \approx x\_{(k+1)\Delta t} + (\sigma_{(k+1)\Delta t}^2 - \sigma_{k\Delta t}^2)\nabla_{x\_{(k+1)\Delta t}}\log p_{k+1}(x\_{(k+1)\Delta t}|y) +\sqrt{\sigma_{(k+1)\Delta t}^2 - \sigma_{k\Delta t}^2}z$ where $z\sim \mathcal{N}(0,I)$. From (31), (32) and the argument in our manuscript, when $\Delta t\rightarrow 0$, we have $p(x\_{(k+1)\Delta t}|y) = \mathcal{N}(E[x_0|y],\sigma\_{(k+1)\Delta t}^2I+C\_{x\_0})$, i.e., $\nabla_{x_{(k+1)\Delta t}}\log p(x_{(k+1)\Delta t}|y)= (\sigma_{(k+1)\Delta t}^2I+C_{x_0})^{-1}(E[x_0|y]-x_{(k+1)\Delta t})$.
> > >
> > > Finally, we have $x_{k\Delta t} = x_{(k+1)\Delta t} + (\sigma_{(k+1)\Delta t}^2I+C_{x_0})^{-1}(\sigma_{(k+1)\Delta t}^2 - \sigma_{k\Delta t}^2)(E[x_0|y]-x_{(k+1)\Delta t}) +\sqrt{\sigma_{(k+1)\Delta t}^2 - \sigma_{k\Delta t}^2}z$, which can be simplify as $x_{k } = x_{k+1} + (\sigma_{k+1 }^2I+C_{x_0})^{-1}(\sigma_{ k+1 }^2 - \sigma_{k}^2)(E[x_0|y]-x_{k+1}) +\sqrt{\sigma_{k+1}^2 - \sigma_{k}^2}z$.
> > >
> > > Thus, the mean is given by the $x_{k+1} + (\sigma_{k+1 }^2I+C_{x_0})^{-1}(\sigma_{ k+1 }^2 - \sigma_{k}^2)(E[x_0|y]-x_{k+1})$ which is the same as (34) in our manuscript. The covariance is given by $(\sigma_{k+1}^2 - \sigma_{k}^2)I$, which is equivalent to the covariance in equation (34) of our manuscript when $\Delta t \rightarrow 0$, as $\sigma_{k+1}^2 \rightarrow \sigma_{k}^2$. We note that the variance here is a scalar, which justifies the fixed precision updated approximation in our paper.
> > >
> > > On the other hand, from the Tweedie formula: $\nabla_{x_{(k+1)\Delta t}}\log p_{k+1}(x_{(k+1)\Delta t}|y) = \frac{1}{\sigma_{(k+1)\Delta t}^2}(E[x_0|x_{(k+1)\Delta t)},y]-x_{(k+1)\Delta t})$, we have $x_{k \Delta t} \approx x_{(k+1)\Delta t} +  \frac{\sigma_{(k+1)\Delta t}^2 - \sigma_{k\Delta t}^2}{\sigma_{(k+1)\Delta t}^2}(E[x_0|x_{(k+1)\Delta t)},y]-x_{(k+1)\Delta t}) + \sqrt{\sigma_{(k+1)\Delta t}^2 - \sigma_{k\Delta t}^2}z $, which is a different Gaussian. The two Gaussian are equivalent when $\Delta t\rightarrow 0$.
> > >
> > > The Gaussian derived from the Tweedie formula can be incorporated into the RMP framework, as we can calculate $E[x_k|x_{k+1},y]$ using the Tweedie formula, which can further be computed through score functions using stochastic NGD proposed in our paper.

---

> > > > ### Comment · Reviewer_k8qn · 2024-11-28
> > > >
> > > > I would like to sincerely thank the authors for the clear justification, and I believe that these valuable discussions are worth being included in the camera-ready version of the paper. Hope to see this paper being accepted!

---

### Official Review · Reviewer_YouM · 2024-11-04

**Soundness:** 2
**Presentation:** 2
**Contribution:** 1
**Rating:** 3
**Confidence:** 4

**Summary:**

The authors propose Reverse Mean Propagation (RMP) as a new framework that leverages score-based variational inference to solve inverse problems by tracking the posterior mean at each step of the reverse diffusion process. RMP is claimed to be more computationally efficient than traditional diffusion-based methods that rely on generating multiple samples from the posterior, as it can achieve the posterior mean directly.

**Strengths:**

The topic of solving inverse problems with score-based models is interesting.

**Weaknesses:**

* The posterior mean is not the only relevant quantity; access to
  samples from the posterior distribution also enables uncertainty
  quantification. This work provides only an approximation to the mean,
  sacrificing sample generation for computational efficiency, which may
  not even be fully realized unless unjustified approximations are made
  (see more below).

* While RMP aims to reduce complexity relative to sampling methods, it still requires nested optimization loops and numerous neural network evaluations.

* The method relies heavily on approximating the likelihood score, $\nabla_{x_k} \log p(y | x_k)$, which is acknowledged as "hard to handle in general." The chosen approximation—replacing the likelihood score with the score at the MMSE estimate of $x_0$—introduces an error quantified by Jensen's Gap.


* The performance of RMP depends on multiple hyperparameters, such as step sizes and the likelihood score balancing parameter. Sensitivity to these hyperparameters and their generalizability across datasets and inverse problems remain unclear.

* Algorithm 2 uses a fixed-precision update to avoid computing the Hessian further introducing approximation errors.

* The authors note that the reverse conditional probability is strictly Gaussian only as $\Delta t \to 0$, an assumption that may not hold in practice, introducing further errors.

* In high-dimensional settings, computing the MMSE estimate of $x_0$ at each step, as required for likelihood score approximation, can be computationally intensive. The manuscript lacks a detailed analysis of RMP’s computational cost in high-dimensional scenarios.

**Questions:**

See above

---

> ### Author Response · Authors · 2024-11-23
>
> **We thank the reviewer for careful reading and constructive comments. We summarize our work as follows:**
>
> The Fokker-Planck equation [1,2] describes the evolution of the probability density of a stochastic process which is deterministic. In this work, we present an evolution analysis of the probability density for the conditional reverse diffusion process, as described by the Fokker-Planck equation. Based on the analysis, we propose a variational inference framework that minimizes the reverse KL divergence, referred to as RMP, since it propagates the mean at each reverse step. Different from the previous works that draw samples from the posterior with a reverse diffusion process, our proposed RMP algorithm is essentially an approximation of the Fokker-Planck equation for a conditional reverse process that tracks the deterministic evolution of the probability density. This represents a fundamental difference between our method and previous methods. Unlike sampling-based methods that have to sample from the posterior multiple times and average out to get the posterior mean which is time-consuming, RMP converges to the posterior mean.
>
> The main contributions of our work are summarized as follows:
>
> - We characterize the evolution of the transaction probability density of the conditional reverse diffusion process in terms of its mean and covariance. Based on this characterization, we propose the RMP framework, which approaches the posterior mean.
> - We connect the variational inference with conditional diffusion process and propose an implementation of the RMP framework based on stochastic natural gradient descent with score-based prior and suitable approximation that reduces complexity.
> - We conduct extensive experiments to demonstrate the validity of the theory of RMP framework and shows that RMP outperforms state-of-the-art algorithms on reconstruction performance with lower computational complexity in various problems.
>
> [1] Risken, H. "The Fokker-Planck Equation." (1996)
>
> [2] Jordan, Richard, David Kinderlehrer, and Felix Otto. "The variational formulation of the Fokker-Planck equation." *SIAM journal on mathematical analysis* 29.1 (1998): 1-17.
>
> **Reply to reviewer's comments:**
>
> - W1: While RMP aims to reduce complexity relative to sampling methods, it still requires nested optimization loops and numerous neural network evaluations.
>
>   > This is not an issue with RMP. For example, in our image reconstruction experiments, we set the inner loop number of VP-RMP to 1, resulting in an algorithm without nested loops. The comparison of VP-RMP with existing sampling-based methods, such as DPS, DDRM, and PnP-ADMM, demonstrates that VP-RMP outperforms these methods with fewer NFEs and reduced running time.
>
> - W2: The method relies heavily on approximating the likelihood score, ∇𝑥𝑘log⁡𝑝(𝑦|𝑥𝑘), which is acknowledged as "hard to handle in general." The chosen approximation—replacing the likelihood score with the score at the MMSE estimate of 𝑥0—introduces an error quantified by Jensen's Gap.
>
> > There are many existing methods for approximating likelihood scores, and the approximation introduced by DPS is just one of them. In Appendix E.3, we compare our approach with the likelihood approximation methods introduced by DPS and PiGDM, demonstrating that RMP performs well with these approximations.
> - W3: The performance of RMP depends on multiple hyperparameters, such as step sizes and the likelihood score balancing parameter. Sensitivity to these hyperparameters and their generalizability across datasets and inverse problems remain unclear.
>
>   > We perform a parameter sensitivity analysis for the hyperparameters (stepsize s1 and 𝛾𝑘) involved in the RMP algorithm, as detailed in the Appendix. Figures 5 and 6 in Appendix E.1 present the performance of VP-RMP and VE-RMP concerning these hyperparameters, measured by four metrics, in the form of a 2D colormesh. From the colormesh, we observe that all four metrics show similar variations and change smoothly with respect to the hyperparameters. Therefore, we can select the hyperparameters based on the colormesh, as shown in Tables 3 and 4.
>
> - W4: Algorithm 2 uses a fixed-precision update to avoid computing the Hessian further introducing approximation errors.
>
>   > Our proposed RMP is the first method to approximate the evolution of the density of the conditional reverse diffusion process. Unlike previous methods that sample from the posterior distribution of data using a SDE, which is a stochastic sampling method, our method tracks the density evolution, which is deterministic. It true that the fixed-precision update to avoid computing complexity introduces approximation error when T is small. We show that the approximation involved in fixed-precision update is robust to many applications as shown in our experiments of Gaussian mixture model and various image reconstruction tasks.

---

> > ### Author Response · Authors · 2024-11-23
> >
> > - The authors note that the reverse conditional probability is strictly Gaussian only as Δ𝑡→0, an assumption that may not hold in practice, introducing further errors.
> >
> >   > In this work, we present an evolution analysis of the transition probability density of the conditional reverse diffusion process, as described by the Fokker-Planck equation. Different from the previous works that draw samples from the posterior distribution of data using a SDE, our proposed RMP algorithm is essentially a score-based implementation of the Fokker-Planck equation for a conditional reverse process that tracks the evolution of the probability density of that process. The implementation, based on variational inference, serves as an approximation method for the discretization of the evolution process. The approximation error for finite T is generally an open question since p(x0|y) is generally intractable. The RMP algorithm presented in Algorithm 2 is a discrete realization of the RMP framework, meaning that RMP approaches the posterior mean strictly as T tends to infinity. The theoretical performance of RMP when the T is finite can be characterized by the variational loss. We next give an intuition on the theoretical bound of approximation error. From the discrete variational loss F defined in (49), a continuous loss can also be defined similarly. The theoretical performance of RMP when T is finite can be characterized by the variational loss. Next, we will provide an intuition regarding the theoretical bound on the approximation error. Similarly, from the discrete variational loss F defined in (49), a continuous loss can also be defined. The difference between the discrete loss and the continuous loss can be interpreted as the approximation error bound. The continuous loss can be upper bounded by the discrete loss using Jensen's inequality, and Jensen's gap also provides the approximation bound. We have added an experiment in the appendix of our manuscript to demonstrate the performance trend of VP-RMP compared to the outer loop number T, as shown in Fig. 7.
> >
> > - In high-dimensional settings, computing the MMSE estimate of 𝑥0 at each step, as required for likelihood score approximation, can be computationally intensive. The manuscript lacks a detailed analysis of RMP’s computational cost in high-dimensional scenarios.
> >
> >   > We have conducted a complexity analysis for general settings and compared the complexity of RMP with existing algorithms like DPS, DDRM, PnP-ADMM for various high-dimensional image reconstruction tasks in our experiment.

---

> > > ### Comment · Reviewer_YouM · 2024-11-24
> > > **Authors did not address my concern regarding motivation**
> > >
> > > While authors provide responses regarding most of my comments, the following issue that I raised remains unresolved:
> > >
> > > > The posterior mean is not the only relevant quantity; access to samples from the posterior distribution also enables uncertainty quantification. This work provides only an approximation to the mean, sacrificing sample generation for computational efficiency, which may not even be fully realized unless unjustified approximations are made (see more below).
> > >
> > > The core issue is that the authors sacrifice guarantees and uncertainty quantification purely for the sake of computational speed. While their approximations (like using a fixed Hessian estimate) do accelerate the calculation of conditional means, this efficiency comes without justification---a concern echoed across multiple reviews---and only based on empirical evidence. The method delivers only point estimates, abandoning the valuable uncertainty information that posterior sampling would provide. This tradeoff seems particularly unfortunate for real-world applications, where having a slightly less precise estimate with reliable uncertainty bounds is often more valuable than a potentially better point estimate without any guarantees. Based on this fundamental limitation, I stand by my original score.

---

> > > > ### Author Response · Authors · 2024-11-24
> > > > **Further clarification of the motivation**
> > > >
> > > > We agree with the reviewer that tracking the posterior mean provides a point estimate of the problem. But an inverse problem, by definition, is a point estimate problem, since it requires the exact reconstruction of the original data. So, we do not understand what is the need to pursue "uncertainty information that posterior sampling would provide". In fact, replacing random sampling by tracking the "deterministic" probability density evolution is exactly the key innovation of our proposed method. As clearly evidenced by our experiments, our "deterministic" approach provides significantly better reconstruction performance than the existing random sampling approaches.
> > > >
> > > > It is true that, just like the existing random sampling approaches, some approximations are made in the realisation of our proposed method. Admittedly, it is difficult to provide theoretical guarantee for variational inference methods. So far, we only rely on the experimental results to demonstrate the effectiveness of our method. We plan to upload our source code after the acceptance of the paper, to ensure the reproducibility of our results by the community.

---

### Comment · Area_Chair_Yzu9 · 2024-11-26

Dear Reviewers ZpSm, nsm7, 4zYw,
If not already, could you please take a look at the authors' rebuttal? Thank you for this important service.
-AC

---

### Meta-Review · Area_Chair_Yzu9 · 2024-12-19

**Metareview:**

This paper proposes a variational inference method for score-based approach to Bayesian inverse problems. The main contribution is a reverse mean propogation method (RMP) that directly approximates the posterior mean, which is implemented by solving multiple variational problems via natural gradient descent. While reviewers and I agree this is an interesting idea, major concerns included compromised accuracy due to numerous approximations and (more thorough) comparison to existing approaches and they seemed to have not completed resolved. Recognizing the potential of the idea, I recommend the authors take the discussions into consideration and submit again.

**Additional Comments On Reviewer Discussion:**

Additional observation: YouM was the main negative reviewer. He did not increase score after rebuttal. On the other hand, the positive reviewers all came from the same group.

---

### Decision · Program_Chairs · 2025-01-22

Reject